# Grey wolf genomic history reveals a dual ancestry of dogs

Anders Bergström[1,69 ✉], David W. G. Stanton[2,3,4,69], Ulrike H. Taron[5,69], Laurent Frantz[4,6], Mikkel-Holger S. Sinding[7,8,9,10], Erik Ersmark[2,3], Saskia Pfrengle[11,12], Molly Cassatt-Johnstone[13], Ophélie Lebrasseur[14], Linus Girdland-Flink[15,16], Daniel M. Fernandes[17,18], Morgane Ollivier[19], Leo Speidel[1,20], Shyam Gopalakrishnan[7], Michael V. Westbury[5,7], Jazmin Ramos-Madrigal[7], Tatiana R. Feuerborn[7,9,11], Ella Reiter[11], Joscha Gretzinger[11,21], Susanne C. Münzel[11], Pooja Swali[1], Nicholas J. Conard[22,23], Christian Carøe[7], James Haile[14], Anna Linderholm[3,14,24,25], Semyon Androsov[26], Ian Barnes[27], Chris Baumann[23,28], Norbert Benecke[29], Hervé Bocherens[23,30], Selina Brace[27], Ruth F. Carden[31], Dorothée G. Drucker[23], Sergey Fedorov[32], Mihály Gasparik[33], Mietje Germonpré[34], Semyon Grigoriev[32,70], Pam Groves[35], Stefan T. Hertwig[36,37], Varvara V. Ivanova[38], Luc Janssens[39], Richard P. Jennings[16], Aleksei K. Kasparov[40], Irina V. Kirillova[41], Islam Kurmaniyazov[42], Yaroslav V. Kuzmin[43], Pavel A. Kosintsev[44], Martina Lázničková-Galetová[45], Charlotte Leduc[46], Pavel Nikolskiy[47], Marc Nussbaumer[36], Cóilín O'Drisceoil[48], Ludovic Orlando[49], Alan Outram[50], Elena Y. Pavlova[51], Angela R. Perri[52,53], Małgorzata Pilot[54], Vladimir V. Pitulko[40], Valerii V. Plotnikov[55], Albert V. Protopopov[55], André Rehazek[36], Mikhail Sablin[56], Andaine Seguin-Orlando[49], Jan Storå[57], Christian Verjux[58], Victor F. Zaibert[59], Grant Zazula[60,61], Philippe Crombé[62], Anders J. Hansen[7], Eske Willerslev[7,63], Jennifer A. Leonard[64], Anders Götherström[3,57], Ron Pinhasi[17,68], Verena J. Schuenemann[11,12,17], Michael Hofreiter[5], M. Thomas P. Gilbert[7,65], Beth Shapiro[13,66], Greger Larson[14], Johannes Krause[67], Love Dalén[2,3] & Pontus Skoglund[1 ✉]

The grey wolf (*Canis lupus*) was the first species to give rise to a domestic population, and they remained widespread throughout the last Ice Age when many other large mammal species went extinct. Little is known, however, about the history and possible extinction of past wolf populations or when and where the wolf progenitors of the present-day dog lineage (*Canis familiaris*) lived[1–8]. Here we analysed 72 ancient wolf genomes spanning the last 100,000 years from Europe, Siberia and North America. We found that wolf populations were highly connected throughout the Late Pleistocene, with levels of differentiation an order of magnitude lower than they are today. This population connectivity allowed us to detect natural selection across the time series, including rapid fixation of mutations in the gene *IFT88* 40,000–30,000 years ago. We show that dogs are overall more closely related to ancient wolves from eastern Eurasia than to those from western Eurasia, suggesting a domestication process in the east. However, we also found that dogs in the Near East and Africa derive up to half of their ancestry from a distinct population related to modern southwest Eurasian wolves, reflecting either an independent domestication process or admixture from local wolves. None of the analysed ancient wolf genomes is a direct match for either of these dog ancestries, meaning that the exact progenitor populations remain to be located.

The grey wolf (*Canis lupus*) has been present across most of the northern hemisphere for the last few hundred thousand years and, unlike many other large mammals, did not go extinct in the Late Pleistocene. Studies of present-day genomes have found that current population structure formed mostly in the last ~30,000–20,000 years[9–11], or roughly since the Last Glacial Maximum (LGM; ~28–23 thousand years ago (ka)[12]). Siberian wolves predating the LGM have ancestries that are largely basal to present-day diversity, which has led to suggestions that many pre-LGM wolf lineages went extinct[13,14]. Among the central questions is thus to what extent the global wolf population was subject to extinction processes or responded to climate change with new adaptations.

While it is clear that grey wolves gave rise to dogs, there is no consensus regarding when, where and how this happened[1–8]. Skeletal remains attributable to the present-day dog lineage appear archaeologically by 14 ka[15], and genetic estimates of when the ancestors of dogs and modern wolves diverged range from 40–14 ka[9,13,16]. However, genetic

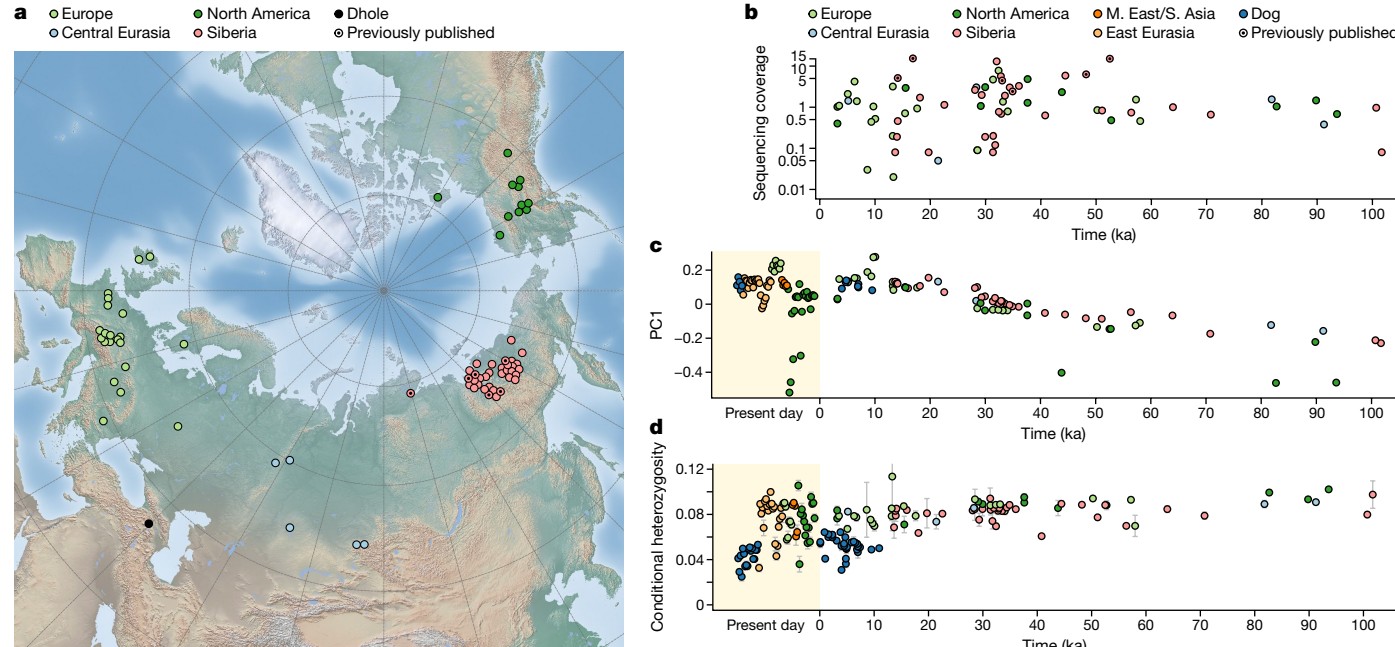

**Fig. 1 | Seventy-two ancient wolf genomes. a**, Sampling locations of ancient wolves and one ancient dhole analysed here, on a base map from Natural Earth (naturalearthdata.com). **b**, Ages and sequencing coverage of ancient wolves. **c**, PC1 from a PCA on outgroup $f_3$-statistics plotted against sample age. PCs were calculated from ancient wolves only, with present-day wolves and dogs projected onto the plot. **d**, Heterozygosity estimates from sampling of two reads at sites ascertained as heterozygous in a coyote. Bars denote 95% CIs from block jackknifing.

data from modern and ancient dogs coupled with modern wolves, to which previous studies were largely restricted, may not be able to resolve the origin of dogs. Genetic diversity within dogs is affected by their dynamic history and is unable to confidently pinpoint an origin. Relationships to modern wolves can likewise be affected by local extinction and gene flow since domestication[6,9]. Regions where early dogs have been found do not necessarily imply places of origin either, as the existence of earlier dogs elsewhere cannot be excluded. Instead, the origin of dogs could be resolved if wolf genetic diversity across space and time was exhaustively characterized and it could be determined which populations were closest to the ancestors of dogs.

## Wolf genomes spanning 100,000 years

We sequenced 66 new ancient wolf genomes from Europe, Siberia and north-western North America to a median of 1× coverage (range, 0.02–13×) (Fig. 1a,b), incorporated five previously sequenced ancient wolf genomes[14,17] and increased coverage for one[13]. We also sequenced an ancient dhole genome from the Caucasus, contextually dated to >70 ka, to serve as an outgroup. Fractions of X-chromosome DNA showed that 69% of the wolves were male (95% confidence interval (CI), 57–80%; $P = 0.0013$, binomial test), mirroring male over-representation among ancient genomes from woolly mammoths[18], bison[19], brown bears[19] and domestic dogs[8]. For wolves without dates or with dates beyond the radiocarbon limit of ~50 ka, we estimated ages through mitochondrial tip dating[20] and obtained an average 95% CI of 21,573 years and an average prediction error of 5,133 years (Supplementary Figs. 1 and 2). We merged single-nucleotide polymorphism (SNP) genotypes called from these genomes with those from worldwide modern wolves ($n = 68$), modern ($n = 369$) and ancient ($n = 33$) dogs, and other canid species (Methods). The total dataset spans the last 100,000 years (Fig. 1b).

In a principal component analysis (PCA) on a matrix of shared genetic drift, the ancient wolves clustered strongly by age and not by geography (Pearson's $r_{PC1,sample\ age} = 0.85$, $P = 5 \times 10^{-21}$) (Fig. 1c). Similarly, ancient wolves share more drift with modern wolves the younger they are (Extended Data Fig. 1a and Supplementary Fig. 3). Previous studies

have suggested an LGM ancestry turnover[13,14,21], and, indeed, we found that all individuals younger than the LGM (that is, postdating 23 ka) were more similar to each other than to wolves predating ~28 ka (Extended Data Fig. 1b). However, the same pattern is also visible when contrasting affinities to younger versus older wolves at any point during the last 100,000 years (Supplementary Fig. 4). Using simulations, we confirmed that the observed temporal relationships are largely similar to what would be expected in a panmictic population (Supplementary Fig. 5). A long-standing process of ancestry homogenization due to connectivity thus seems to have driven Pleistocene wolf relationships. The changes during the LGM therefore represent not a shift in long-term population dynamics, but the most recent manifestation of this process.

## Siberia as a source of global gene flow

We next tested for directionality in the gene flow that connected wolf ancestry over time. Analyses using $f_4$-statistics showed that all wolves postdating 23 ka are more similar to Siberian wolves than to European or Central Asian wolves from ~30 ka (Extended Data Fig. 1c and Supplementary Fig. 6). This suggests that Siberian-related ancestry expanded into Europe, in line with mitochondrial evidence[21]. The same dynamic of Siberian gene flow into Europe unfolded between 50 and 35 ka (Supplementary Fig. 6). We found that an admixture graph model with recurrent, unidirectional gene flow from Siberia into Europe could explain these relationships (Fig. 2a and Supplementary Fig. 8). Although we could not distinguish pulse-like from continuous gene flow, our results suggest that Siberia acted as a source and Europe as a sink for migration throughout the Late Pleistocene and show no evidence of gene flow in the other direction (Extended Data Fig. 1d and Supplementary Fig. 7).

While these results demonstrate pervasive gene flow, they also show that the ancestry replacements were incomplete and that minority fractions of deep European ancestry have persisted until the present day (Fig. 2a,b). Most analysed modern Eurasian wolves probably retain local Pleistocene ancestry, as they are best modelled by qpAdm as having 10–40% ancestry that is more divergent than the oldest Siberian wolves in this study at ~100 ka (Supplementary Figs. 11 and 12). In addition to

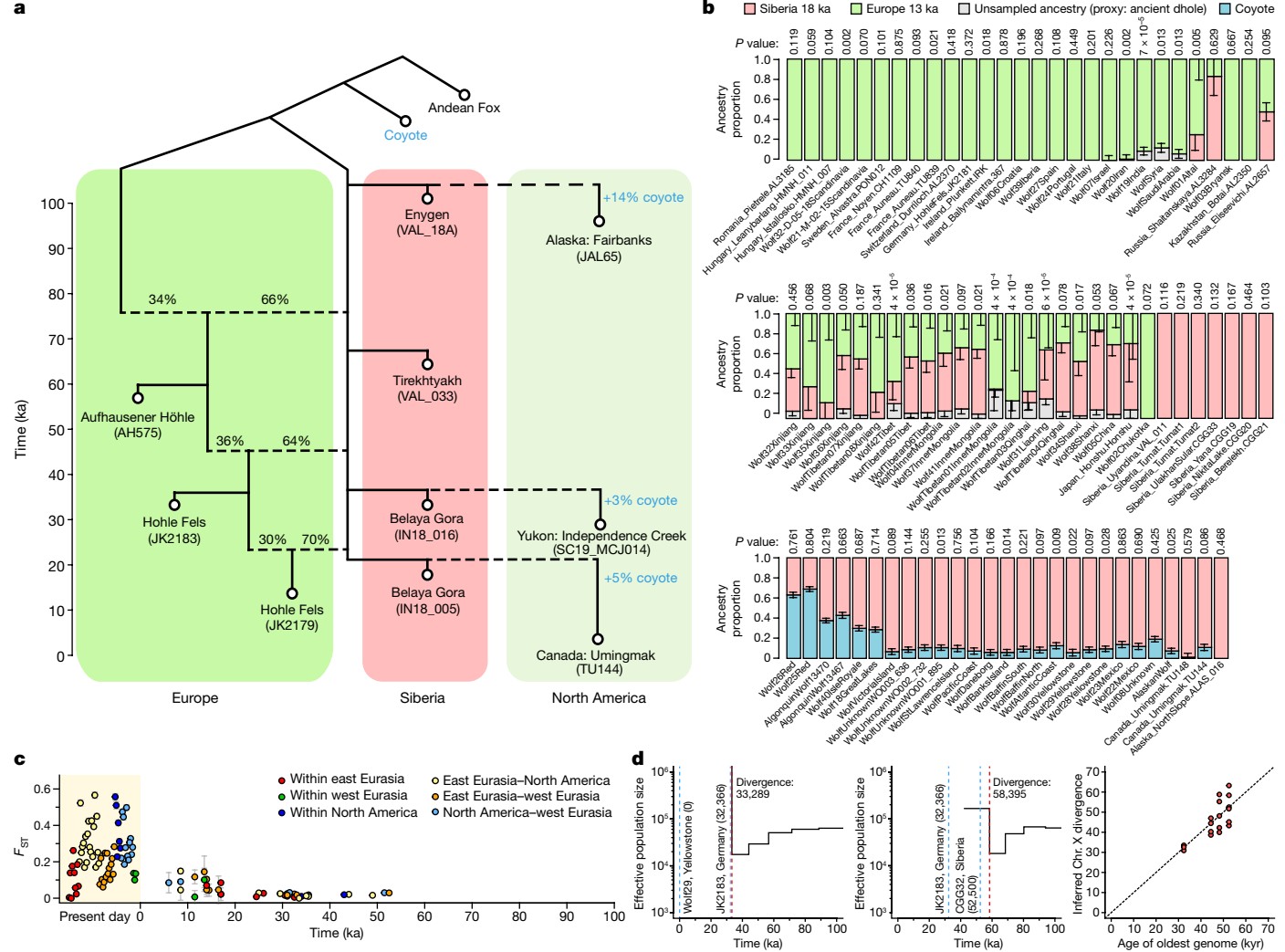

**Fig. 2 | One hundred thousand years of wolf population history. a**, Admixture graph fit by qpGraph to selected ancient wolves, with two outlier (|*Z*| > 3) *f*-statistics (worst = 3.16). **b**, Best-fitting qpAdm models for post-LGM and present-day wolves. An ancient dhole was used as the outgroup for Eurasian wolves to capture any unsampled divergent ancestry, while a coyote was used as the outgroup for North American wolves. Bars denote ±1 standard error estimated from a block jackknife. **c**, $F_{ST}$ for pairs of sample groups with mean dates separated by ≤12,500 years. Bars denote ±1.96 standard errors **d**, MSMC2 results for pairs of male X chromosomes, with sample ages indicated by blue lines. A sharp upwards spike in the curve corresponds to population divergence, with estimated timings indicated by red lines. Example curves for two pairs of wolves (left and middle) and a summary of results for all pairs (right) are shown. kyr, thousand years.

local grey wolf ancestry not represented among our ancient genomes, this may include African golden wolf-related ancestry in the Near East and South Asia[22] and ancestry of unknown canid origin in Tibet[23] (Supplementary Fig. 10). While all Eurasian wolves today share the majority of their ancestry within the last 25,000 years, the persistence of deep local ancestries provides evidence against widespread local extinction in Late Pleistocene Eurasia and suggests that the species as a whole, unlike many other megafauna, did not come close to extinction.

Many modern and ancient North American wolves show evidence of coyote (*Canis latrans*) admixture[24,25] (Extended Data Fig. 1e), which explains why some of them do not cluster with wolves of similar age in the PCA (Fig. 1c). On the basis of coalescence rates[26] between male X chromosomes, which have perfect haplotype phase, we estimated that wolves and coyotes began diverging ~700 ka (Supplementary Fig. 14), broadly in line with a fossil divergence of ~1 million years ago[27]. Our data show that coyote admixture has occurred at least since 100–80 ka, and two analysed Pleistocene wolves from the Yukon also carried coyote mitochondrial lineages. These findings imply that either the Pleistocene range of coyotes extended further north than currently thought

or that admixture occurring further south propagated northwards through the wolf population. In our Eurasian wolves, no influx of coyote ancestry is observed over time (Extended Data Fig. 1e). We found a slight west–east gradient of increasing coyote affinity among Eurasian wolves, but this pattern probably reflects admixture into coyotes from North American wolves (which are related to wolves in eastern Siberia) (Supplementary Fig. 9).

After accounting for coyote admixture, we found that wolf ancestry in Alaska and the Yukon was highly connected to Siberia over time (Fig. 2a). This mirrors European wolf history, but, while some deep local European ancestry persists, no deep North American ancestry appears to persist to the present. The Bering land bridge probably allowed for an influx of Siberian wolves into Alaska intermittently between 70 and 11 ka[28,29], but we found no evidence of gene flow in the other direction. All present-day North American wolves can be modelled as having 10–20% coyote ancestry and the remaining ancestry from Siberian wolves younger than ~23 ka, with no contribution from earlier North American wolves (Fig. 2b). We found that red and Algonquin wolves similarly fit as shifted towards coyotes along this two-source admixture

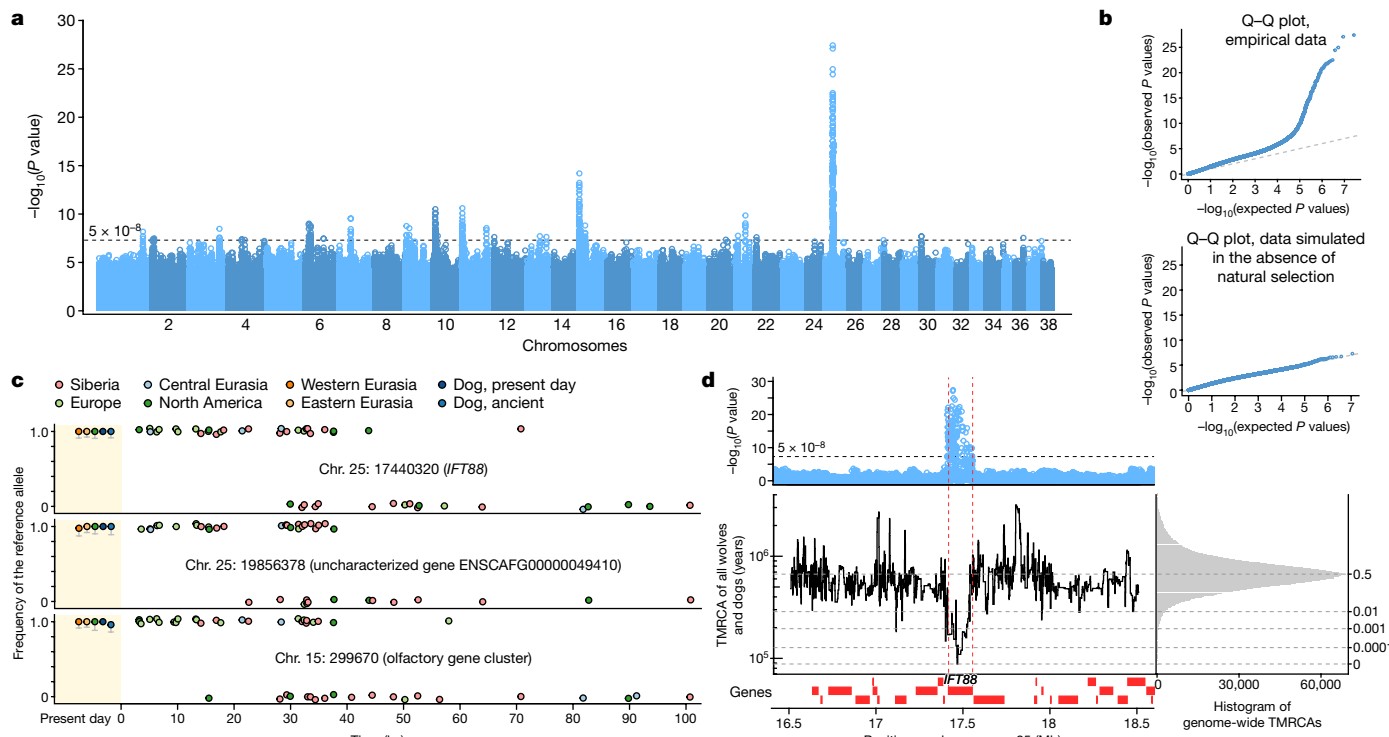

**Fig. 3 | Natural selection in the ancient wolf time series. a**, $-\log_{10}(P$ values) (two sided, not adjusted for multiple comparisons) from linear regression for association between allele frequency and sample age. **b**, Quantile–quantile plot comparing the $P$ values in **a** to those expected under a uniform distribution (top) and likewise for results from a simulated neutrally evolving population (effective population size ($N_e$) = 50,000) (bottom). **c**, Allele observations in ancient wolves and frequencies in present-day populations for lead variants from the three strongest peaks. Bars denote 95% binomial CIs. **d**, Local $P$ values (from **a**) and TMRCA inferred using Relate on modern wolves and dogs for the region surrounding *IFT88*. The genome-wide histogram (quantiles in grey lines) shows that this locus has the most recent TMRCA in the genome.

cline[11,25], but we cannot rule out greater complexity in their history. While genomic data alone cannot establish an absence of grey wolves at any particular time, our results are consistent with local extinction in North America, for example during the LGM when ice sheets covered the northern half of the continent[30], or, alternatively, an absence of grey wolves south of the ice sheets until after the ice retreated.

## High connectivity in the Pleistocene

To understand how differentiated past wolf populations were, we calculated the proportion of genetic variation between rather than within (pairwise $F_{ST}$; ref.[31]) sets of wolves grouped in space and time. Before the LGM, differentiation even between distant regions was low ($F_{ST} < 3\%$) (Fig. 2c). Early European and North American populations were thus neither very different from each other nor from the Siberian-related wolves that over time replaced much of their ancestry. We also estimated X-chromosome coalescence rates[26], which suggested that any two Pleistocene wolves shared ancestry within ~10,000 years of the date of the older wolf (Fig. 2d and Supplementary Fig. 15). Pervasive gene flow thus prevented deep divergences among wolf populations in the Late Pleistocene.

In the last ~10,000 years (the Holocene), population dynamics were different from those in the Pleistocene, with no evidence for further Siberian gene flow into Europe; instead, European-related ancestry spread eastwards and contributed to modern wolves in China and Siberia (Fig. 2b). Higher levels of differentiation today ($F_{ST}$ of ~10–60%) probably largely reflect population bottlenecks following habitat encroachment and persecution by humans in the last few centuries[32,33], although there is some evidence for increasing differentiation already during the last 20,000 years (Fig. 2c). MSMC2 estimates from present-day

genomes suggest widespread effective population size declines in this period (Supplementary Fig. 13), but we found no concurrent decline in individual heterozygosity (Fig. 1d). Combined, this evidence suggests that an overall reduction in gene flow, as shown by the $F_{ST}$ results, rather than a species-wide population decline[21] might have resulted in lower local effective population sizes.

## Natural selection over 100,000 years

The strong connectivity observed among Late Pleistocene wolves raises the possibility of species-wide adaptation. Natural selection is typically inferred indirectly from present-day genetic variation, but our 100,000-year (~30,000 generations) dataset enables direct detection of selected alleles. Testing each variant for an association between allele frequency and time across 72 ancient and 68 modern wolves, and applying genomic control[34] to correct for allele frequency variance caused by genetic drift, we found 24 genomic regions with evidence for selection (Fig. 3a and Extended Data Table 1). We confirmed the robustness of our method to demographic history by applying it to data simulated in the absence of selection, finding no false positives (Fig. 3b and Supplementary Fig. 17).

The strongest signal was observed on chromosome 25, where variants closely overlapping the gene *IFT88* rose rapidly from close to 0% to 100% in frequency 40–30 ka and are still fixed in wolves and dogs today (Fig. 3c). Genealogical inference on modern wolves[35,36] further showed that *IFT88* had the youngest time to the most recent common ancestor (TMRCA) (~70,000 years) in the genome (Fig. 3d). Disruption of *IFT88* leads to craniofacial development defects in mice and to cleft lip and palate in humans[37]. If future fossil studies reveal rapid craniodental change in this time period, this could implicate the *IFT88* sweep as a

driver, potentially in response to prey availability changes. But it is also possible that selection targeted unknown non-skeletal traits associated with *IFT88* variation. The second strongest signal in the genome was 2.5 Mb downstream of *IFT88*, where allele frequencies shifted in a similar timeframe 40–20 ka (Fig. 3c), but it is not clear whether this region could be involved in long-range regulation of *IFT88*.

Three regions with evidence for selection overlap olfactory receptor genes, with variants on chromosome 15 increasing in frequency from close to 0% to 100% 45–25 ka (Fig. 3c), suggesting that olfaction was a recurrent target of adaptation in wolves. Most of the detected selection episodes occurred before the divergence of dogs, and dogs share the selected alleles (Supplementary Fig. 18). However, variants in *YME1L1* increased in frequency from <5% to 50–70% in wolves from 20–0 ka but are not observed in dogs. A region on chromosome 10, where variation among dogs is associated with body size, drop ears and other traits[38–40], was under recent selection in specific dog breeds[41], and we found that it was also selected in wolves in the last 20,000 years. Although it was not detected in our selection scan, the $K^B$ deletion that underlies black fur[42] was identified in a 14,000-year-old wolf from Tumat, Siberia (Supplementary Fig. 19). This deletion probably introgressed into wolves from dogs in the Holocene[42], but our result also raises the possibility that its ultimate origin could have been in wild Pleistocene wolves.

## Dog ancestry has eastern wolf affinities

We found that dogs share more genetic drift with wolves that lived after 28 ka than with those that lived before this time, which implies that the progenitors of dogs were genetically connected to other wolves at least until 28 ka (Fig. 1c and Extended Data Fig. 1b). A divergence around this time is also consistent with our MSMC2 analyses of X chromosomes (Supplementary Fig. 16). However, until the nature of the divergence process is better understood, it cannot be ruled out that domestication had started before this point.

The geographical origin of the present-day dog lineage *Canis familiaris* has remained controversial. Genetic studies have argued that wolves in East Asia[1,2], Central Asia[4], the Middle East[6], Europe[5], Siberia[16], or both eastern and western Eurasia independently[3], contributed ancestry to early dogs, whereas others have been consistent with a single, but geographically unknown, progenitor population[8,9]. Given our finding that part of wolf population structure is older than the likely time of dog domestication, we can expect dogs to be genetically closer to some ancient wolves than to others. To reduce the effects of gene flow since the emergence of dogs, we performed a PCA on wolves and dogs from the last 25,000 years, based on $f_4$-statistics quantifying their relationships only to wolves living before 28 ka (that is, before the LGM), and found that dogs showed relationship profiles similar to those of Siberian wolves from 23–13 ka (Fig. 4a, Extended Data Fig. 2 and Methods). Direct $f_4$-tests also showed that dogs are closer to Siberian than to European wolves from this period (Fig. 4b and Extended Data Fig. 3). European wolves postdating 28 ka have an affinity to pre-LGM European wolves, reflecting the persistence of deep west Eurasian wolf ancestry (Fig. 2a). The absence of such western affinities in dogs suggests that they did not originate from the European wolf populations sampled here.

While the north-eastern Siberian wolves from 23–13 ka display the greatest overall affinity to dogs, we found that they were not the immediate ancestors of dogs. When a broad set of ancient wolves were tested as candidate sources using qpWave/qpAdm[43], all single-source models, including one using an 18,000-year-old Siberian wolf, were strongly rejected for all dogs studied ($P < 1 \times 10^{-6}$) (Methods and Fig. 4c). However, a model featuring the Siberian wolf and 10–20% ancestry from a component approximated by the outgroup dhole fit dogs such as the 9,500-year-old Siberian Zhokhov[17] individual ($P = 0.29$) (Fig. 4c). Although it uses an outgroup species, this two-source model does not necessarily imply admixture from two distinct populations or

species. Instead, it could reflect dogs being derived from some local wolf ancestry that is unsampled and to some extent divergent from the available ancient wolves (Extended Data Fig. 4). Validating this interpretation, we found that recent European wolves, which have a small degree of deep, local European ancestry (Fig. 2a), obtain results very similar to those for dogs, requiring 10–20% unsampled ancestry, if only Siberian wolves were available as sources (Supplementary Fig. 11 and Supplementary Information). We therefore interpret the results for dogs as similarly reflecting some unsampled wolf ancestry that is not fully represented by the ancient Siberian wolves sampled here. This unsampled ancestry appears to have retained a partial degree of differentiation from the sampled ancient wolves since before 100 ka (Supplementary Fig. 12), and our results imply that it probably lived outside the regions of Europe, north-eastern Siberia and North America sampled here.

The results obtained for the Zhokhov dog also applied to ancient dogs from Lake Baikal, North America and north-eastern Europe (a 10,900-year-old Karelian dog) and to modern New Guinea singing dogs. As a group, qpWave could fit these dogs as having originated from a single 'stream' of ancient wolf diversity, in an approach not requiring a proximate source (Extended Data Table 2). This result shows that ancient wolf genomes can circumvent the complexities of more recent processes, as the same models were rejected when modern wolves were used as sources instead (Extended Data Table 2), probably owing to gene flow from dogs into wolves[8].

Recent admixture and population changes thus complicate analyses of modern wolves. Even so, if wolf population structure has not been completely reshaped since the time of dog domestication, it is possible that part of the ancestry of the dog progenitors could still be represented and detectable among wolves today, even though the past geographical location of that ancestry would be unknown. We tested this in two ways. First, we projected dogs onto a PCA plot constructed using modern wolf genotypes, and found that they projected closer to wolves from China, Mongolia and the Altai than to wolves from Yakutia (Extended Data Fig. 5). Second, we extended our qpAdm analyses to modern wolf sources, and found that some Chinese wolves provided better fits than the 18,000-year-old Siberian wolf and could serve as single sources of Zhokhov dog ancestry without the need for an unsampled ancestry component (Extended Data Fig. 6). These results could be taken to support an eastern or central Eurasian dog origin outside of north-eastern Siberia, but we cannot draw firm geographical conclusions in the absence of ancient wolf genomes from these and other candidate regions.

## A second source for western dog ancestry

We extended our analyses to a global set of ancient and modern dogs, to test for any ancestry contributions from additional, genetically distinct wolf progenitors. The strongest evidence for multiple progenitors would be if some dogs had different affinities to wolves that predate domestication, as such wolves cannot be affected by dog gene flow. Applying this rationale, we found that ancient Near Eastern and present-day African dogs, and to a lesser degree European dogs, are shifted towards western Eurasian wolves in the $f_4$-statistics PCA based on relationships to wolves that predate the LGM (Fig. 4a). This cline recapitulates the primary axis of population structure within dogs (between ancient Near Eastern and eastern Eurasian dogs[8]) (Fig. 4b), even when wolves from the last 28,000 years are excluded (Supplementary Fig. 20). The dog ancestry cline thus at least in part reflects wolf ancestry differences that predate the likely domestication timeframe. Testing the PCA observations explicitly, qpWave strongly rejected a single wolf progenitor when including Near Eastern dogs ($P < 10^{-4}$) (Extended Data Table 2). The best-fitting qpAdm models for these dogs instead involved a source related to ancient European wolves, in addition to the ancestry found in the Zhokhov dog (Fig. 4c).

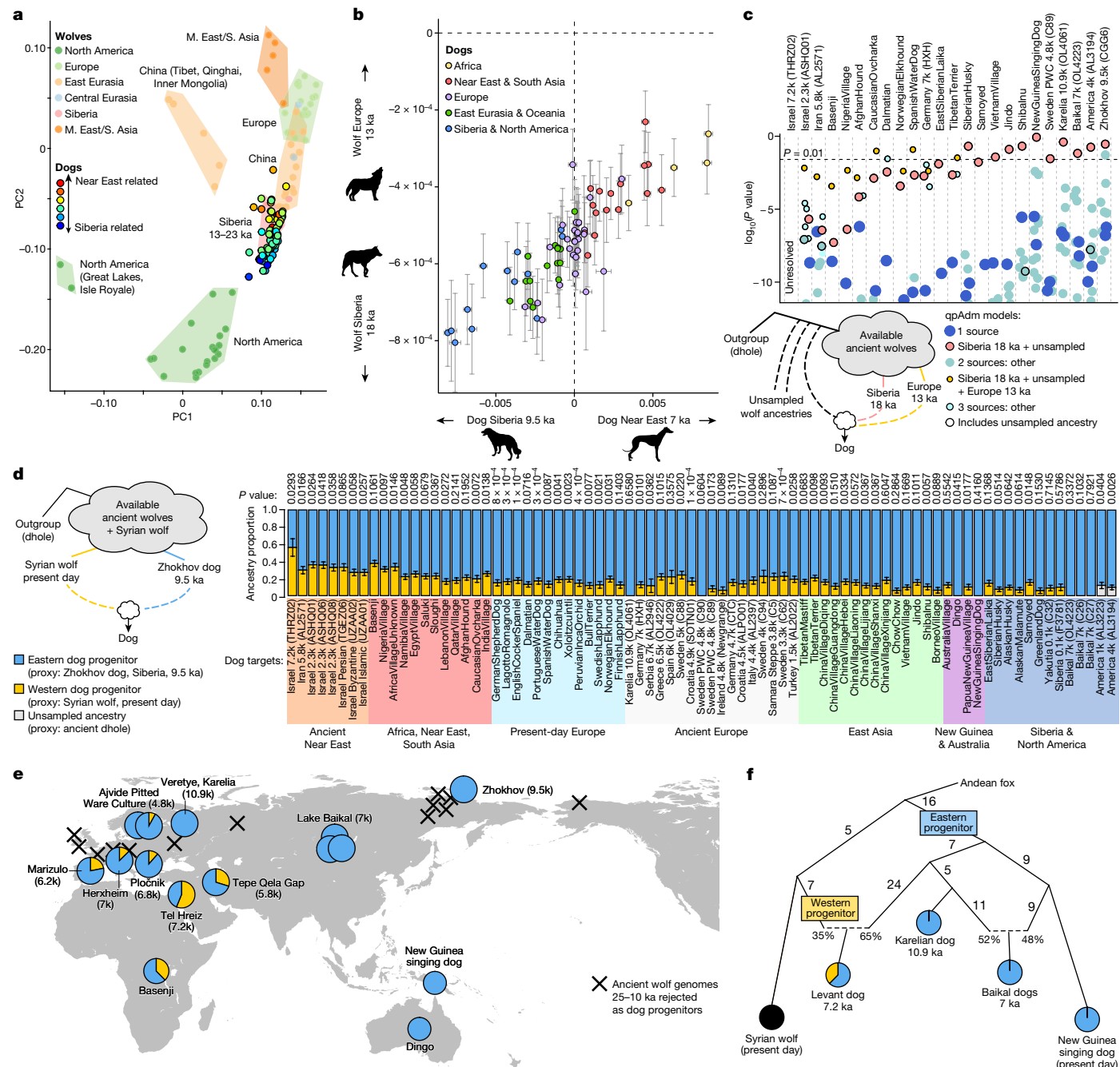

**Fig. 4 | The ancestry of dogs. a**, PCA on post-LGM and present-day wolves (X), based on $f_4$-statistics only of the form $f_4$(X,A;B,C), where A, B and C are any of 21 wolves predating 28 ka. Dogs are projected, and coloured by $f_4$(AndeanFox,X; Zhokhov dog 9.5 ka,Tel Hreiz dog 7.2 ka). **b**, For dogs (X), $f_4$(AndeanFox,X; Zhokhov dog 9.5 ka,Tel Hreiz dog 7.2 ka) horizontally against $f_4$(AndeanFox,X; Belaya Gora wolf 18 ka,Hohle Fels wolf 13 ka) vertically (Pearson's $r = 0.86$, $P = 3 \times 10^{-23}$). Bars denote ±1 standard error estimated from a block jackknife. Silhouettes from phylopic.org. **c**, $\log_{10}(P$ values) for qpAdm models fit to dog targets, where a low $P$ value means the model can be rejected. An ancient dhole was used to represent unsampled, divergent ancestry; models including this

source have black outlines. Points are jittered horizontally to avoid overlap. **d**, qpAdm ancestry proportions for dogs, using the Zhokhov (9.5 ka) dog and a present-day Syrian wolf as proxies for eastern and western dog progenitor ancestry, respectively. Bars denote ±1 standard error estimated from a block jackknife. **e**, Map of early and relevant later dogs and their ancestry proportions as in **d**. Black crosses indicate the locations of wolves from 25–10 ka that can be rejected as dog progenitors. Base map from the mapdata R package. k, thousand years. **f**, Admixture graph model of major dog lineage relationships, fit by qpGraph with no outlier $f$-statistics. Edge lengths are in units of $F_{ST}$ (×1,000).

To test whether the sampled ancient European wolves could be the actual source of this second component of dog ancestry, we tested qpAdm models featuring the Siberian Zhokhov dog as one source—representing the eastern-related dog ancestry—and an ancient European wolf as a second source. These models did not fit

Near Eastern and African dogs unless a third, outgroup component was also included to represent unsampled, divergent ancestry (Supplementary Fig. 21), meaning that European wolves are not a match for the missing ancestry. Expanding to all post-LGM and present-day wolves, only present-day wolves from Syria, Israel, Iran and India

achieved good fits (Extended Data Fig. 7). In line with a source from this part of the world, when projected onto present-day wolf structure, Near Eastern and African dogs are shifted towards Caucasian and Near Eastern rather than European wolves (Extended Data Fig. 5). Using a present-day Syrian wolf as a source, we estimated 56% (standard error, 10%) Near Eastern-related wolf ancestry in the earliest available dog (7.2 ka) from the Levant, 37% (standard error, 3.5%) in the African Basenji breed and 5–25% in Neolithic and later European dogs (Fig. 4d). While the evidence of dual ancestry is based on ancient wolves that predate domestication and are thus unaffected by potential later gene flow, these exact estimates could be inflated if there is dog admixture in the Syrian wolf.

Next, we exhaustively tested admixture graph models of dog relationships, allowing up to two admixture events among four dog populations and the Syrian wolf. We obtained results consistent with the qpAdm inferences, as a single graph featuring Syrian wolf admixture into early Near Eastern dogs fit the data (Fig. 4f), with a separate dog lineage giving rise to early Karelian and eastern dogs. In this graph, the Karelian dog is most closely related to the 'eastern' source that also contributed ancestry to the early Near Eastern dog.

The widespread ancestry asymmetries observed between wolves and dogs today have been interpreted as reflecting recent, local admixture[8,9]. Our finding that dogs have variable proportions of two distinct components of wolf ancestry may provide a unifying explanation for many of these asymmetries. For example, previous studies have explained an affinity between Pleistocene Siberian wolves and Arctic dogs by suggesting admixture in the latter[13,17]. The dual ancestry model can probably explain this asymmetry without such admixture, with the Arctic dogs instead having less of the western component (Supplementary Fig. 22). Conversely, higher levels of the western component in Near Eastern and African dogs probably explains at least part of their previously observed affinity to Near Eastern wolves[8,9,10]. An observation that wolves in Xinjiang, central Asia, display no asymmetries to different dogs was interpreted as suggesting that other asymmetries are primarily due to dog-to-wolf gene flow[8]. Our results instead suggest that a balance of eastern and western wolf ancestries in central Asia (Fig. 2b) causes relative symmetry to the eastern and western dog ancestries. The Xinjiang wolves are thus not evidence against the dual ancestry model.

## Conclusion

We show that wolf populations were genetically connected throughout the Late Pleistocene, probably because of the high mobility of wolves in an open landscape[44]. The LGM did not necessarily correspond to an unprecedented time of change for the interconnected population of wolves, which might provide a clue to their perseverance when other northern Eurasian carnivores became extinct. Furthermore, the reason Pleistocene wolves appear basal to present-day diversity is not that they went extinct[13,14], but that continued gene flow homogenized later ancestry. Our finding that several selected alleles quickly reached fixation shows that adaptations spread to the whole population of Pleistocene wolves, a process that might have contributed to the survival of the species. At the same time, our results show that such rapid species-wide selective sweeps occurred only a few times over the last ~100,000 years.

Our results also provide insights into long-standing questions on the origin of dogs. First, dogs and present-day Eurasian wolves have been thought to be reciprocally monophyletic lineages[9]. We find that, overall, dogs are closer to eastern Eurasian wolves. Second, because no modern wolves are a good match for dog ancestry, the source population has been assumed to be extinct. Our results imply that this is not necessarily the case, as continued homogenization of wolf ancestry could have obscured earlier relationships to dogs. Third, it has been unclear whether more than one wolf population contributed to early and present-day dogs[3,7,8,9]. We find that an eastern Eurasian-related source, 'eastern dog progenitor', appears to have contributed ~100% of the ancestry of early dogs in Siberia, the Americas, East Asia and north-eastern Europe. On top of this, a western Eurasian-related source, 'western dog progenitor', contributed 20–60% of the ancestry of early Near Eastern and African dogs and 5–25% of the ancestry of Neolithic and later European dogs. The western ancestry subsequently spread worldwide with, for example, the prehistoric expansion of agriculture in western Eurasia[8] and the colonial era expansion of European dogs.

A previous study proposed that the earlier archaeological appearance of dogs in western and eastern Eurasia than in central Eurasia was due to independent domestication of western and eastern wolves, but that ancestry from the former was extinct or nearly extinct in present-day dogs[3]. Our results support the notion of two distinct ancestors of dogs but differ from this previous hypothesis. First, we demonstrate that ancestry from at least two wolf populations is extant and ubiquitous in modern dogs, and is the major determinant of dog population structure today. Second, we are able to reject Pleistocene European wolves related to those sampled here as a source for the *C. familiaris* lineage. Third, the previous study suggested that an Irish Neolithic dog had more ancestry from the western domestication than later dogs[3], whereas we find that this dog had less ancestry from the western progenitor identified here than present-day European dogs (Fig. 4d). The lack of genomes from the earliest dogs in Europe, however, means that future studies may reveal them to have arisen from an independent domestication process that did not contribute substantially to later populations[3,45,46].

Our results are consistent with two scenarios: (1) independent domestication of the eastern and western progenitors that later merged in the west or (2) single domestication of the eastern progenitor, followed by admixture from western wolves as dogs arrived into southwestern Eurasia. Our results cannot distinguish between these scenarios, but, in either case, the merging or admixture must have occurred before 7.2 ka, the age of the oldest available Near Eastern dog genome[8]. A single domestication of the western progenitor followed by admixture from eastern wolves does not seem compatible with our results, as it would require replacement of 100% of the ancestry of eastern dogs. If dogs of 100% western progenitor ancestry were discovered, for example, in the earliest Near Eastern[47] or European[15] contexts, this would imply independent domestication. Alternatively, the first dogs in the west could be of eastern progenitor ancestry, similar to the Karelian dog from 10.9 ka, in line with a single domestication process. Additional ancient wolf genomes, including from outside the regions covered here, where DNA often preserves less well, will also be necessary to further identify the wolf progenitors of dogs.

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

¹Ancient Genomics Laboratory, The Francis Crick Institute, London, UK. ²Department of Bioinformatics and Genetics, Swedish Museum of Natural History, Stockholm, Sweden. ³Centre for Palaeogenetics, Stockholm, Sweden. ⁴School of Biological and Behavioural Sciences, Queen Mary University of London, London, UK. ⁵Evolutionary Adaptive Genomics, Institute of Biochemistry and Biology, University of Potsdam, Potsdam, Germany. ⁶Palaeogenomics Group, Department of Veterinary Sciences, Ludwig Maximilian University, Munich, Germany. ⁷The GLOBE Institute, University of Copenhagen, Copenhagen, Denmark. ⁸Smurfit Institute of Genetics, Trinity College Dublin, Dublin, Ireland. ⁹The Qimmeq Project, University of Greenland, Nuuk, Greenland. ¹⁰Greenland Institute of Natural Resources, Nuuk, Greenland. ¹¹Institute for Archaeological Sciences, University of Tübingen, Tübingen, Germany. ¹²Institute of Evolutionary Medicine, University of Zurich, Zurich, Switzerland. ¹³Department of Ecology and Evolutionary Biology, University of California, Santa Cruz, Santa Cruz, CA, USA. ¹⁴The Palaeogenomics & Bio-Archaeology Research Network, Research Laboratory for Archaeology and History of Art, University of Oxford, Oxford, UK. ¹⁵Department of Archaeology, School of Geosciences, University of Aberdeen, Aberdeen, UK. ¹⁶School of Biological and Environmental Sciences, Liverpool John Moores University, Liverpool, UK. ¹⁷Department of Evolutionary Anthropology, University of Vienna, Vienna, Austria. ¹⁸CIAS, Department of Life Sciences, University of Coimbra, Coimbra, Portugal. ¹⁹University of Rennes, CNRS, ECOBIO (Ecosystèmes, biodiversité, évolution)–UMR 6553, Rennes, France. ²⁰Genetics Institute, University College London, London, UK. ²¹Max Planck Institute for the Science of Human History, Jena, Germany. ²²Department of Early Prehistory and Quaternary Ecology, University of Tübingen, Tübingen, Germany. ²³Senckenberg Centre for Human Evolution and Palaeoenvironment, University of Tübingen, Tübingen, Germany. ²⁴Texas A&M University, College Station, TX, USA. ²⁵Department of Geological Sciences, Stockholm University, Stockholm, Sweden. ²⁶Museum 'Severnyi Mir', Yakutsk, Russian Federation. ²⁷Department of Earth Sciences, Natural History Museum, London, UK. ²⁸Department of Geosciences and Geography, Faculty of Science, University of Helsinki, Helsinki, Finland. ²⁹German Archaeological Institute, Berlin, Germany. ³⁰Biogeology, Department of Geosciences, University of Tübingen, Tübingen, Germany. ³¹School of Archaeology, University College Dublin, Dublin, Ireland. ³²North-Eastern Federal University, Yakutsk, Russian Federation. ³³Hungarian Natural History Museum, Budapest, Hungary. ³⁴Royal Belgian Institute of Natural Sciences, Brussels, Belgium. ³⁵University of Alaska, Fairbanks, AK, USA. ³⁶Naturhistorisches Museum Bern, Bern, Switzerland. ³⁷Institute of Ecology and Evolution, University of Bern, Bern, Switzerland. ³⁸VNIIOkeangeologiya, St Petersburg, Russian Federation. ³⁹University of Leiden, Leiden, the Netherlands. ⁴⁰Institute for the History of Material Culture, Russian Academy of Sciences, St Petersburg, Russian Federation. ⁴¹Ice Age Museum, Shidlovskiy National Alliance 'Ice Age', Moscow, Russian Federation. ⁴²Department of Archaeology, Ethnology and Museology, Al-Farabi Kazakh State University, Almaty, Kazakhstan. ⁴³Sobolev Institute of Geology and Mineralogy, Siberian Branch of the Russian Academy of Sciences, Novosibirsk, Russian Federation. ⁴⁴Ural Federal University, Yekaterinburg, Russian Federation. ⁴⁵Moravian Museum, Brno, Czech Republic. ⁴⁶INRAP, Metz, France. ⁴⁷Geological Institute, Russian Academy of Sciences, Moscow, Russian Federation. ⁴⁸National Monuments Service, Department of Housing, Local Government and Heritage, Dublin, Ireland. ⁴⁹Centre d'Anthropobiologie et de Génomique de Toulouse UMR 5288, CNRS, Faculté de Médecine Purpan, Université Paul Sabatier, Toulouse, France. ⁵⁰Department of Archaeology, University of Exeter, Exeter, UK. ⁵¹Arctic & Antarctic Research Institute, St Petersburg, Russian Federation. ⁵²PaleoWest, Henderson, NV, USA. ⁵³Department of Anthropology, University of Nevada, Las Vegas, Las Vegas, NV, USA. ⁵⁴Museum & Institute of Zoology, Polish Academy of Sciences, Gdańsk, Poland. ⁵⁵Academy of Sciences of Sakha Republic, Yakutsk, Russian Federation. ⁵⁶Zoological Institute of the Russian Academy of Sciences, St. Petersburg, Russian Federation. ⁵⁷Stockholm University, Stockholm, Sweden. ⁵⁸Service Régional de l'Archéologie, Orléans, France. ⁵⁹Institute of Archaeology and Steppe Civilizations, Al-Farabi Kazakh National University, Almaty, Kazakhstan. ⁶⁰Yukon Palaeontology Program, Whitehorse, Yukon Territories, Canada. ⁶¹Collections and Research, Canadian Museum of Nature, Ottawa, Ontario, Canada. ⁶²Department of Archaeology, Ghent University, Ghent, Belgium. ⁶³Department of Zoology, University of Cambridge, Cambridge, UK. ⁶⁴Estación Biológica de Doñana (EBD-CSIC), Sevilla, Spain. ⁶⁵University Museum, NTNU, Trondheim, Norway. ⁶⁶Howard Hughes Medical Institute, University of California, Santa Cruz, Santa Cruz, CA, USA. ⁶⁷Max Planck Institute for Evolutionary Anthropology, Leipzig, Germany. ⁶⁸Human Evolution and Archaeological Sciences, University of Vienna, Vienna, Austria. ⁶⁹These authors contributed equally: Anders Bergström, David W. G. Stanton, Ulrike H. Taron. ⁷⁰Deceased: Semyon Grigoriev. ✉e-mail: anders.bergstrom@crick.ac.uk; pontus.skoglund@crick.ac.uk

## Methods

### Sampling, DNA preparation and sequencing

**Stockholm.** Samples LOW002, LOW003, LOW006, LOW007, LOW008 and PON012 were processed at the Archaeological Research Laboratory at Stockholm University, Sweden, following methods previously described[8]. In brief, this involved extracting DNA by incubating the bone powder for 24 h at 37 °C in 1.5 ml of digestion buffer (0.45 M EDTA (pH 8.0) and 0.25 mg ml$^{-1}$ proteinase K), concentrating supernatant on Amicon Ultra-4 (30-kDa molecular weight cut-off (MWCO)) filter columns (Merck-Millipore) and purifying on Qiagen MinElute columns. Double-stranded Illumina libraries were prepared using the protocol outlined in ref. [48], with the inclusion of USER enzyme and the modifications described in ref. [49].

Samples 367, PDM100, Taimyr-1 and Yana-1 were processed at the Swedish Museum of Natural History in Stockholm, Sweden, following previously described methods[8]. In brief, this involved extracting DNA using a silica-based method with concentration on Vivaspin filters (Sartorius), according to a protocol optimized for recovery of ancient DNA[50]. Double-stranded Illumina libraries were prepared using the protocol outlined in ref. [48], with the inclusion of USER enzyme.

Samples ALAS_024, VAL_033, ALAS_016, VAL_008, HMNH_007, HMNH_011, VAL_050, VAL_005, DS04, VAL_037, VAL_012, VAL_011, VAL_18A, IN18_016 and IN18_005 were processed at the Swedish Museum of Natural History in Stockholm, Sweden, following previously described methods for permafrost bone and tooth samples[51]. In brief, this involved DNA extraction using the methodology of ref. [52] and double-stranded Illumina library preparation as described in ref. [48], with dual unique indexes and the inclusion of USER enzyme. Between eight and ten separate PCR reactions with unique indexes were carried out for each sample to maximize library complexity. The libraries were sequenced alongside samples HOV4, AL2242, AL2370, AL2893, AL3272 and AL3284 across three Illumina NovaSeq 6000 lanes with an S4 100-bp paired-end set-up at SciLifeLab in Stockholm.

**Potsdam.** Samples JAL48, JAL65, JAL69, JAL358, AH574, AH575 and AH577 were processed at the University of Potsdam. Pre-amplification steps (DNA extraction and library preparation) were conducted in separated laboratory rooms specially equipped for the processing of ancient DNA. Amplification and post-amplification steps were performed in different laboratory rooms. DNA was extracted from bone powder (29–54 mg) following a protocol specially adapted to recover short DNA fragments[52]. Single-stranded double-indexed libraries were built from 20 µl of DNA extract according to the protocol in ref. [53]. The libraries were sequenced on an HiSeq X platform at SciLifeLab in Stockholm.

**Tübingen/Jena.** Samples JK2174, JK2175, JK2179, JK2181, JK2183, TU144, TU148, TU839 and TU840 were processed at the University of Tübingen, with DNA extraction and pre-amplification steps undertaken in clean room facilities and post-amplification steps performed in a separate DNA laboratory. Both laboratories fulfil standards for work with ancient DNA[54,55]. All surfaces of tooth and bone samples were initially UV irradiated for 30 min, to minimize the potential risk of modern DNA contamination. Subsequently, DNA was extracted by applying a well-established guanidine silica-based protocol for ancient samples[52]. Illumina sequencing libraries were prepared by using 20 µl of DNA extract per library[48]; afterwards, dual barcodes (indexes) were chemically added to the prime ends of the libraries[56]. For the samples from Auneau (TU839 and TU840), five sequencing libraries each were prepared; for all other samples processed in Tübingen, three sequencing libraries each were prepared. To detect potential contamination of the chemicals, negative controls were conducted for extraction and library preparation. After preparation of the sequencing libraries, DNA concentration was measured with qPCR (Roche LightCycler) using corresponding primers[48]. The DNA concentration was given by the copy number of the DNA fragments in 1 µl of the sample.

Amplification of the indexed sequencing libraries was performed using Herculase II Fusion under the following conditions: 1× Herculase II buffer, 0.4 µM IS5 primer and 0.4 µM IS6 primer[48], Herculase II Fusion DNA polymerase (Agilent Technologies), 0.25 mM dNTPs (100 mM; 25 mM each dNTP) and 0.5–4 µl barcoded library as template in a total reaction volume of 100 µl. The applied amplification thermal profile was processed as follows: initial denaturation for 2 min at 95 °C; denaturation for 30 s at 95 °C, annealing for 30 s at 60 °C and elongation for 30 s at 72 °C for 3 to 20 cycles; and a final elongation step for 5 min at 72 °C. Thereafter, the amplified DNA was purified using a MinElute purification step and DNA was eluted in 20 µl TET. The concentration of the amplified DNA sequencing libraries was measured using a Bioanalyzer (Agilent Technologies) and a DNA1000 lab chip from Agilent Technologies.

The sequencing libraries were sequenced on an Illumina HiSeq 4000 platform at the Max Planck Institute for Science of Human History in Jena. The samples from Auneau (TU839 and TU840) were paired-end sequenced applying 2 × 50 + 8 + 8 cycles. All other libraries prepared in Tübingen were single-end sequenced using 75 + 8 + 8 cycles.

**Oxford.** Samples AL2657, AL2541, AL2741, AL2744, AL3185, AL2350, CH1109, AL2370, AL3272 and AL3284 were processed at the dedicated ancient DNA facility at the PalaeoBARN laboratory at the University of Oxford, following methods described previously[8]. In brief, double-stranded libraries were constructed following the protocol in ref. [48]. These libraries were sequenced on a HiSeq 2500 (AL2657, AL2541, AL2741, AL2744) or a HiSeq 4000 (AL3185, AL2350, CH1109) instrument at the Danish National Sequencing Center or on a NextSeq 550 instrument (AL2741) at the Natural History Museum of London. For samples AL2370, AL3272 and AL3284, between six and eight separate PCR reactions with unique indexes were carried out on their libraries and they were sequenced alongside samples HOV4, VAL_18A and IN18_016 on an Illumina NovaSeq 6000 lane with an S4 100-bp paired-end set-up at SciLifeLab in Stockholm.

**Copenhagen.** Samples CGG13, CGG17, CGG19, CGG20, CGG21, CGG25, CGG26, CGG27, CGG28, CGG34, Tumat1 and IRK were processed at the GLOBE Institute, University of Copenhagen. All pre-PCR work was performed in ancient DNA facilities following ancient DNA guidelines[57]. The details of extraction, library construction and sequencing for the samples with CGG codes are described in ref. [21], in relation to the publication of mitochondrial data from these specimens. The Tumat1 sample was processed following the exact same protocol. In brief, DNA extraction was performed using a buffer containing urea, EDTA and proteinase K[50], double-stranded libraries were prepared with NEBNext DNA Sample Prep Master Mix Set 2 (E6070S, New England Biolabs) and Illumina-specific adaptors[48], and sequencing was performed on an Illumina HiSeq 2500 platform using 100-bp single-read chemistry. For the IRK sample, DNA was extracted from three subsamples and purified as described in ref. [21]. The three DNA extracts and the purified pre-digest of one subsample were incorporated into double-stranded libraries following the BEST protocol[58], with the modifications described in ref. [59], and sequenced on a BGISEQ-500 platform using 100-bp single-read chemistry.

**Santa Cruz.** Samples SC19.MCJ017, SC19.MCJ015, SC19.MCJ010 and SC19.MCJ014 were processed at the UCSC Paleogenomics Lab and were provided by the Yukon Government Paleontology program. All pre-PCR work was performed in a dedicated ancient DNA facility at the University of California, Santa Cruz, following standard ancient DNA methods[60]. Subsamples (250–350 mg) were sent to the UCI KECK AMS facility for radiocarbon dating, and the remaining amounts were powdered in a Retsch MM400 for extraction. For each sample, ~100 mg of powder was treated with a 0.5% sodium hypochlorite solution before extraction to remove surface contaminants[61] and then combined with

1 ml lysis buffer for extraction, following the protocol in ref. [52]. Samples were processed in parallel with a negative control. We quantified the extracts using a Qubit 1× dsDNA HS Assay kit (Q33231) before preparing libraries. We prepared single-stranded libraries following the protocol in ref. [62] and amplified the libraries for 9–16 cycles as informed by qPCR. After amplification, we cleaned the libraries using a 1.2× SPRI bead solution and pooled them to an equimolar ratio for in-house shallow quality-control sequencing on a NextSeq 550 paired-end 75-bp run. We then sent the libraries to Fulgent Genetics for deeper sequencing on two paired-end 150-bp lanes on a HiSeq X instrument.

**Vienna.** Sample HOV4 was processed at the Department of Anthropology, University of Vienna. The sample is a canine tooth, which after sequencing was determined to derive from a dhole (*Cuon alpinus*). DNA was extracted from its cementum using the methods described in ref. [63] with a modified incubation time of ~18 h. The library was prepared according to the protocol in ref. [48] with the modifications from ref. [64]. Five separate PCR reactions with unique indexes were carried out on the library and were sequenced alongside samples VAL_18A, IN18_016, AL2242, AL2370, AL2893, AL3272 and AL3284 on an Illumina NovaSeq 6000 lane with an S4 100-bp paired-end set-up at SciLifeLab in Stockholm.

An overview of all samples and their associated metadata is available in Supplementary Data 1.

## Genome sequence data processing

For paired-end data, read pairs were merged and adaptors were trimmed using SeqPrep (https://github.com/jstjohn/SeqPrep), discarding reads that could not be successfully merged. Reads were mapped to the dog reference genome canFam3.1 using BWA aln (v.0.7.17)[65] with permissive parameters, including a disabled seed (-l 16500 -n 0.01 -o 2). Duplicates were removed by keeping only one read from any set of reads that had the same orientation, length and start and end coordinates. For sample Taimyr-1, previously published data[13] were merged with newly generated data. Data from samples processed in Copenhagen were processed as described previously[66] except that they were also mapped to canFam3.1. Post-mortem damage was quantified using PMDtools (v0.60)[67] with the '--first' and '--CpG' arguments.

## Genotyping and integration with previously published genomes

To construct a comparative dataset for population genetic analyses, we started from a published variant call set compiling 722 modern dog, wolf and other canid genomes from multiple previous studies (NCBI BioProject accession PRJNA448733)[40]. To this, we added additional modern whole genomes from other studies: 4 African golden wolves and 15 Nigerian village dogs (Genome Sequence Archive (http://gsa.big.ac.cn/), accession PRJCA000335)[68], 12 Scandinavian wolves (European Nucleotide Archive accession PRJEB20635)[69], 9 North American wolves and coyotes (European Nucleotide Archive accession PRJNA496590)[25] and 8 other canids (African hunting dog, dhole, Ethiopian wolf, golden jackal, Middle Eastern grey wolves) (European Nucleotide Archive accession PRJNA494815)[22]. Reads from these genomes were mapped to the dog reference genome using bwa mem (version 0.7.15)[70], marked for duplicates using Picard Tools (v2.21.4) (http://broadinstitute.github.io/picard), genotyped at the sites present in the above dataset using GATK HaplotypeCaller (v3.6)[71] with the '-gt_mode GENOTYPE_GIVEN_ALLELES' argument and then merged into the dataset using bcftools merge (http://www.htslib.org/). The following filters were then applied to sites and genotypes across the full dataset: sites with excess heterozygosity (bcftools fill-tags 'ExcHet' $P$ value $< 1 \times 10^{-6}$) were removed; indel alleles were removed by setting the genotype of any individual carrying such an allele to missing; genotypes at sites with a depth (taken as the sum of the 'AD' VCF fields) less than a third of or more than twice the genome-wide average for the given genome or lower than 5 were set to missing; genotypes containing any allele other than the two highest-frequency alleles at the site were set to missing; allele representation was normalized using bcftools norm; and, finally, sites at which 130 or more individuals had a missing genotype were removed. This resulted in a final dataset of 67.8 million biallelic SNPs. In ancestry analyses (that is, those involving $f$-statistics), modern wolves were treated as individuals while for modern dogs up to four individuals with the highest sequencing coverage from a given breed were used and combined into populations. A list of the modern genomes used in analyses and their associated metadata is included in Supplementary Data 2.

All ancient genomes were assigned pseudo-haploid genotypes on the variant sites in the above dataset using htsbox pileup r345 (https://github.com/lh3/htsbox), requiring a minimum read length of 35 bp ('-l 35'), mapping quality of 20 ('-q 20') and base quality of 30 ('-Q 30'). If an ancient genome carried an allele not present in the dataset, its genotype was set to missing. Previously generated ancient and historical wolf and dog genomes mapped to the dog reference were obtained from the respective publications[3,7,8,13,17,66,72,73] (European Nucleotide Archive study accessions PRJEB7788, PRJEB13070, PRJNA319283, PRJEB22026, PRJNA608847, PRJEB38079, PRJEB39580, PRJEB41490) and genotyped in the same way. A list of the ancient genomes used in analyses and their associated metadata is included in Supplementary Data 2.

## Mitochondrial genome phylogenetic analysis and evolutionary dating

We extracted reads mapped to the mitochondrial genome for the ancient wolf samples using samtools (v1.9)[74]. We called consensus sequences using a 75% threshold, calling any sites with coverage less than 3 as 'N', using Geneious (v9.0.5) and removed any samples with greater than 10% missing data. We included a set of previously published mitochondrial genomes from ancient and modern wolves[5,9,13,21,75–80], which led to a final dataset of 183 individuals (62 [14]C-dated ancient individuals, 24 undated ancient individuals of which 7 had infinite [14]C dates, and 90 modern individuals). We also included three coyote-like sequences as outgroups (from one modern coyote and two ancient wolves with coyote-like mitochondrial sequences: SC19.MCJ015, [14]C dated, and SC19.MCJ017, with an infinite [14]C date). We aligned all sequences using Clustal Omega (v1.2.4)[81]. A Bayesian phylogeny was constructed using BEAST (v1.10.1)[82], with an HKY + I + G substitution model chosen by JModelTest2 (v2.1.10)[83], uncorrelated relaxed log-normal clock and coalescent constant size tree prior. We combined 20 MCMC chains (each run for 200 million iterations), after excluding the first 25% of values as a burn-in. For [14]C-dated samples, we included tip date priors that corresponded to a normal distribution with the same mean and 95% confidence distribution as the [14]C dates. We estimated the ages of undated samples from a prior distribution as follows: (1) for the $n = 24$ ancient samples with no [14]C information, we used a uniform prior of 0 to 1,000,000 years before the present (BP); (2) for the $n = 7$ ancient samples with infinite [14]C dates, we used a uniform prior as in (1), but with the lower limit as the minimum date given by the radiocarbon dating; (3) all $n = 90$ modern samples had already been published previously[21], and the tip date priors for these samples were the same as the uniform priors used in the earlier study (either 0 to 100 or 0 to 500 BP). The mitochondrial consensus sequences for the wolf samples newly reported here (excluding those that were removed because they had too much missing data) are available as Supplementary Data 4.

## $f$-statistics and admixture graphs

$f_3$- and $f_4$-statistics were calculated with ADMIXTOOLS (v5.0)[84], using only transversion sites and with the 'numchrom: 38' argument. To overcome memory limitations when calculating large numbers of $f_4$-statistics, block jackknifing was performed external to ADMIXTOOLS across 225 blocks of 10 Mb in size. Admixture graphs were fit using qpGraph, with arguments 'outpop: NULL', 'useallsnps: NO', 'blgsize: 0.05', 'forcezmode: YES',

'lsqmode: NO', 'diag: 0.0001', 'bigiter: 6', 'hires: YES' and 'lambdascale: 1'. Outgroup $f_3$-statistics were calculated using only sites ascertained to be heterozygous in the CoyoteCalifornia individual.

PCA was performed on outgroup $f_3$-statistics by transforming the values to distances by taking $1 - f_3$ and then running the prcomp R function on the resulting distance matrix. Only ancient wolves were included in the calculation of PCs; present-day wolves and ancient and present-day dogs were then individually projected onto the PCs by re-running the analysis once for each of these individuals independently with that single individual added in and saving its coordinates. To avoid overloading the plot with dogs, only the following dogs were included: Basenji, Boxer, BullTerrier, NewGuineaSingingDog, SiberianHusky, Germany. HXH (7,000 BP), Germany.CTC (4.7 ka), Ireland.Newgrange (4,800 BP), Israel.THRZ02 (7,200 BP), Baikal.OL4223 (6,900 BP), Zhokhov.CGG6 (9,500 BP) and PortauChoix.AL3194 (4,000 BP).

PCA was performed on $f_4$-statistics by transforming the values to pairwise distances by taking $\sqrt{2 \times (1 - r)}$, where $r$ is the Pearson correlation for a given pair of individuals, and then running the ppca function from the pcaMethods (v1.74.0) R package on the resulting distance matrix. For the 'pre-LGM PCA' (Fig. 4a and Extended Data Fig. 2), only all possible $f_4$-statistics of the form $f_4(X,A;B,C)$ were included, where X was the post-25 ka and present-day individuals included in the plot and A, B and C were drawn from a reference set of ancient wolves that lived before 28 ka. For each X, the input was thus a vector of $f_4$-statistics that quantified its relationships to pre-LGM wolves. Only wolves (post-25 ka and present day) were included in the calculation of PCs, and ancient and present-day dogs were then individually projected onto the PCs as described above.

## Heterozygosity and $F_{ST}$ estimates

Conditional heterozygosity was estimated at 1,250,173 transversion sites ascertained to be heterozygous in the CoyoteCalifornia individual, chosen because it is largely an outgroup to wolf diversity. For each individual, exactly two reads were sampled at each of these sites (if available), and the fraction of sites where these two reads displayed different alleles was calculated (alleles other than the two observed in the coyote were ignored). Standard errors were obtained by block jackknifing across the 38 chromosomes.

$F_{ST}$ was calculated with smartpca from the EIGENSOFT (v7.2.1) package[85], using the 'inbreed: YES' option to account for the pseudohaploid genotypes of the ancient genomes (this option was also applied to present-day diploid genomes). $F_{ST}$ was calculated pairwise for pools of at least two genomes, formed from individuals selected for being close in time and space (Supplementary Table 1). A few pairs of individuals showed high similarity indicating possible relatedness, as assessed by comparing read mismatch rates across versus within individuals, and one individual from each of these pairs was excluded from these analyses (JK2174 was excluded because of high similarity to JK2183, TU839 because of high similarity to TU840, and CGG17 because of high similarity to Yana-1). $F_{ST}$ values for pairs of pools with age midpoints separated by less than 12,500 years were included in the plot.

## Divergence time and effective population size analyses with MSMC2

We used MSMC2 (v2.1.2)[26] to infer population divergence times and effective population size histories. Input genotypes for this were called using GATK HaplotypeCaller (v3.6)[86] on ancient and modern genomes with sequencing coverage >5.8×. For divergence time analyses, haploid X chromosomes from two different male genomes were combined and the point at which the inferred effective population size for this 'pseudodiploid' chromosome increased sharply upwards was taken to correspond to a population divergence. Results were scaled using a mutation rate of $0.4 \times 10^{-8}$ mutations per site per generation[13,87] (with a 25% lower rate for X-chromosome analyses) and

a mean generational interval of 3 years[13]. For effective population size inferences, transition variants were ignored and results were scaled using a transversions-only mutation rate inferred from results on modern genomes. For more details on the MSMC2 analyses, see Supplementary Information section 3.

## Selection analyses

Selection analysis was performed using PLINK (v1.90b5.2)[88]. This analysis used the 72 ancient wolf genomes and 68 modern wolf genomes (with the latter including a historical Japanese wolf genome[73] treated as ancient for analysis purposes, with its age set to 200 BP). A list of the genomes used for this analysis is available in Supplementary Data 2 ("Used for selection scan" column). All SNPs, not only transversions, were used for this analysis. The age of each wolf was set as the phenotype, with values of 0 for modern wolves, and the '--linear' argument was used to test for an association between SNP genotypes and age, also applying the '--adjust' argument to correct $P$ values using genomic control. The application of genomic control[34] here aimed to use the magnitude of temporal allele frequency variance observed across the genome to account for what was observed from genetic drift alone given wolf demographic history. Only results for the following sets of sites were retained and included in the Manhattan plot: sites where at least 40 ancient genomes had a genotype call, sites with a minor allele frequency among the ancient wolves of ≥5% and sites that had at least 7 neighbouring sites within a 50-kb window with a $P$ value that was at least 90% as large (on a $\log_{10}$ scale) as the $P$ value of the site itself. The last 'neighbourhood filter' aimed to reduce false positives by requiring similar evidence across multiple nearby sites. As a $P$-value significance cut-off to correct for the genome-wide testing, we used $5 \times 10^{-8}$, which is commonly used in genome-wide association studies in humans and also in dogs[89]. We excluded 15 regions where only a single variant reached significance. A detailed table with the 24 detected regions is available in Supplementary Data 3. To test the robustness of this analysis to false positives arising from genetic drift alone, we applied the same analysis to data from neutral coalescent simulations generated using ms[90] and found no false positives. For more details, see Supplementary Information section 4.

## Ancestry modelling with qpAdm and qpWave

We used the qpAdm and qpWave methods[43] from ADMIXTOOLS (v5.0)[84] to test ancestry models for wolf and dog targets postdating 23 ka. For the primary analyses, we used the following set of candidate source populations (age estimate in brackets, years BP): Armenia_Hovk1. HOV4 (ancient dhole), Siberia_UlakhanSular.LOW008 (70,772), Germany_Aufhausener.AH575 (57,233), Siberia_BungeToll.CGG29 (48,210), Germany_HohleFels.JK2183 (32,366), Siberia_BelayaGora. IN18_016 (32,020), Yukon_QuartzCreek.SC19.MCJ010 (29,943), Altai_Razboinichya.AL2744 (28,345), Siberia_BelayaGora.IN18_005 (18,148) and Germany_HohleFels.JK2179 (13,229). We used a rotating approach in which, for each target, we tested all possible one-, two- and three-source models that could be enumerated from the above set. Individuals from the set that were not used as a source in a given model served as the reference set (or the 'right' population in the qpAdm framework). This means that, in every model, each of the above individuals was always either in the source list or in the reference list. We ranked models on the basis of their $P$ values, but prioritized models with fewer sources using a $P$-value threshold of 0.01: if a simpler model (meaning a model with fewer sources) had a $P$ value above this threshold, it ranked above a more complex model (meaning a model with more sources) regardless of the $P$ value of the latter. We also failed models with inferred ancestry proportions larger than 1.1 or smaller than −0.1. For single-source models, qpWave was run instead of qpAdm. Both programs were run with the 'allsnps: YES' option (without this option, there was very little power to reject models). We describe ancestry assigned to the ancient dhole source (Armenia_Hovk1.HOV4) as 'unsampled' ancestry; note that

this does not imply that such ancestry is of non-wolf origin, only that it is not represented by (that is, diverged early from and lacks shared genetic drift with) the ancient wolf genomes in the reference set.

To test whether any post-23 ka or modern wolf genome available might be a good proxy for the western Eurasian wolf-related ancestry identified in Near Eastern and African dogs, we added the 9,500-year-old Zhokhov dog[17] to the rotating set of candidate source populations. Chosen for its high coverage, early date and easterly location, this makes the assumption that the Zhokhov dog is a good representative for the eastern dog ancestry component. Using the African Basenji dog as a target, models involving the Zhokhov dog plus another given wolf thus allowed us to test whether that wolf was a good match for the additional component of ancestry. For more details on the qpAdm and qpWave analyses, see Supplementary Information sections 2 (wolf targets) and 5 (dog targets).

## Reporting summary

Further information on research design is available in the Nature Research Reporting Summary linked to this paper.

## Data availability

The generated DNA sequencing data are available in the European Nucleotide Archive (ENA) under study accession PRJEB42199. Previously published genomic data analysed here are available under accession numbers PRJNA448733, PRJCA000335, PRJEB20635, PRJNA496590, PRJNA494815, PRJEB7788, PRJEB13070, PRJNA319283, PRJEB22026, PRJNA608847, PRJEB38079, PRJEB39580 and PRJEB41490, with individual genomes used listed in Supplementary Data 2. The canFam3.1 reference genome is available under NCBI assembly accession GCF_000002285.3.

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

**Acknowledgements** This work was supported by grants to P. Skoglund from the European Research Council (grant no. 852558), the Erik Philip Sörensen Foundation and the Science for Life Laboratory, Swedish Biodiversity Program, made available by support from the Knut and Alice Wallenberg Foundation. A.B., L.S., P. Swali and P. Skoglund were supported by Francis Crick Institute core funding (FC001595) from Cancer Research UK, the UK Medical Research Council and the Wellcome Trust. P. Skoglund was also supported by the Vallee Foundation, the European Molecular Biology Organisation and the Wellcome Trust (217223/Z/19/Z). Computations were supported by SNIC-UPPMAX. We also acknowledge support from Science for Life Laboratory, the Knut and Alice Wallenberg Foundation, the National Genomics Infrastructure funded by the Swedish Research Council and the Uppsala Multidisciplinary Center for Advanced Computational Science for assistance with massively parallel sequencing and access to the UPPMAX computational infrastructure. We thank the Yukon gold mining community and First Nations, including the Tr'ondëk Hwëch'in, for continued support of our palaeontology research in the Yukon Territories, Canada. We thank the Danish National High-Throughput Sequencing Centre and BGI-Europe for assistance in sequencing data generation and the Danish National Supercomputer for Life Sciences–Computerome (https://computerome.dtu.dk) for computational resources. We thank National Museum Wales for continued sampling support. M. Germonpré acknowledges support from the Brain.be 2.0 ICHIE project (BELSPO B2/191/P2/ICHIE). M.T.P.G. was supported by the European Research Council (grant no. 681396). M.-H.S.S. was supported by the Velux Foundations through the Qimmeq Project, the Aage og Johanne Louis-Hansens Fond and the Independent Research Fund Denmark (8028-00005B). L.D. acknowledges support from FORMAS (2018-01640). D.W.G.S. received funding for this project from the European Union's Horizon 2020 research and innovation programme under Marie Skłodowska-Curie grant agreement no. 796877. M.P. was supported by the Polish National Agency for Academic Exchange–NAWA (grant no. PPN/PPO/2018/1/00037). V.J.S. was supported by the University of Zurich's University Research Priority Program 'Evolution in Action: From Genomes to Ecosystems'. This research was done with the participation of ZIN RAS (grant no. 075-15-2021-1069). We are grateful to the museum of the Institute of Plant and Animal Ecology UB RAS (Ekaterinburg, Russia) for provision of samples. R.P.J. and C.O'D. were supported by the Standing Committee for Archaeology of the Royal Irish Academy through the Archaeological Excavation Research Grant Scheme. E.Y.P., P.N. and V.V.P. are supported by the Russian Science Foundation (grant no. 16-18-10265-RNF and 21-18-00457-RNF). Y.V.K. was supported by the Russian Science Foundation

(grant no. 20-17-00033). M.H. was supported by the European Research Council (consolidator grant GeneFlow no. 310763). M.L.-G. was supported by the Czech Science Foundation GAČR (grant no. 15-06446S) and institutional financing of the Moravian Museum from the Czech Ministry of Culture (IP DKRVO 2019-2023, MK000094862). L.S. is supported by the Sir Henry Wellcome fellowship (220457/Z/20/Z). We thank Staatliches Museum für Naturkunde Stuttgart for sample access. L.F. and G.L. were supported by European Research Council grants (ERC-2013-StG-337574-UNDEAD and ERC-2019-StG-853272-PALAEOFARM) and Natural Environmental Research Council grants (NE/K005243/1, NE/K003259/1, NE/S007067/1 and NE/S00078X/1). L.F. was also supported by the Wellcome Trust (210119/Z/18/Z). This research was funded in whole, or in part, by the Wellcome Trust (FC001595). For the purpose of open access, the author has applied a CC-BY public copyright licence to any author accepted manuscript version arising from this submission.

**Author contributions** A.J.H., E.W., J.A.L., A.G., R.P., V.J.S., M.H., M.T.P.G., B.S., G.L., J.K., L.D. and P. Skoglund supervised the study. S.A., N.B., H.B., R.F.C., D.G.D., S.F., M. Gasparik, M. Germonpré, S. Grigoriev, P.G., S.T.H., V.V.I., L.J., R.P.J., A.K.K., I.V.K., I.K., Y.V.K., P.A.K., M.L.-G., C.L., P.N., M.N., C.O'D., A.O., E.Y.P., V.V. Pitulko, V.V. Plotnikov, A.V.P., A.R., M.S., J.S., C.V., V.F.Z., G.Z. and P.C. excavated or curated samples. D.W.G.S., U.H.T., M.-H.S.S., E.E., S.P., M.C.-J., O.L., L.G.-F., D.M.F., M.O., M.V.W., T.R.F., E.R., J.G., S.C.M., N.J.C., C.C., J.H., A.L., I.B., C.B., S.B., L.O., A.R.P. and A.S.-O. generated data through sample preparation and/or laboratory work. A.B., D.W.G.S., U.H.T., L.F., M.-H.S.S., L.S., S.G., J.R.-M., P. Swali, M.P. and P. Skoglund analysed and/or curated genomic data. A.B. and P. Skoglund wrote the paper with input from all authors.

**Competing interests** The authors declare no competing interests.

**Additional information**

**Correspondence and requests for materials** should be addressed to Anders Bergström or Pontus Skoglund.

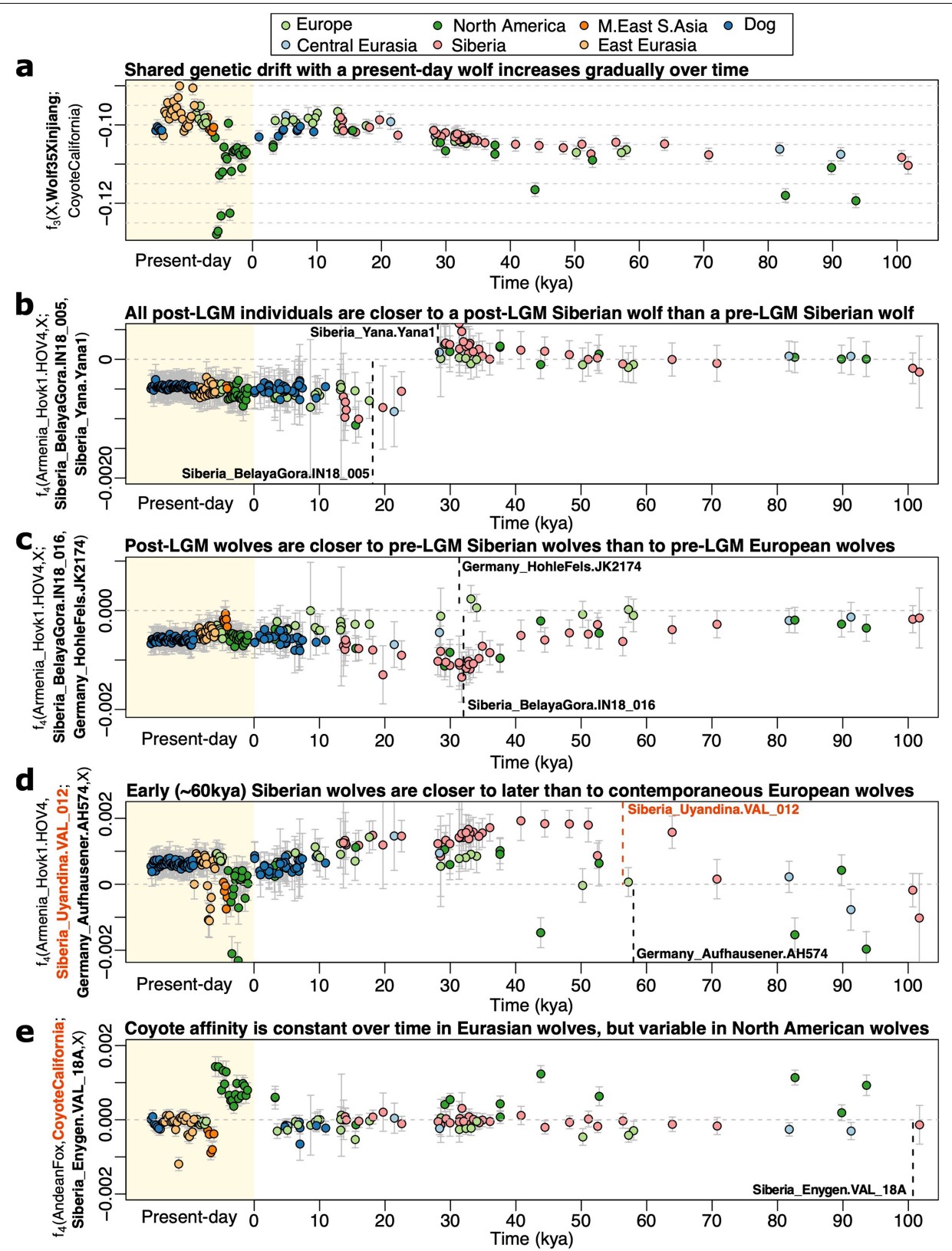

**Extended Data Fig. 1 | f-statistics informing on wolf population history.**
Bars denote ±1.96 standard errors for $f_3$-statistics, and ±3 standard errors for
$f_4$-statistics, estimated from a block jackknife. a) Outgroup $f_3$-statistics
quantifying shared genetic drift with a present-day wolf (Fig. S3). b) $f_4$-statistics
contrasting affinities to a pre-LGM and a post-LGM Siberian wolf (Fig. S4).
c) $f_4$-statistics contrasting affinities to a Siberian and a European pre-LGM wolf
(Fig. S6). d) $f_4$-statistics quantifying whether a ~60 ky old Siberian wolf is closer
to a contemporaneous European wolf or other individuals (Fig. S7).
e) $f_4$-statistics quantifying whether a coyote is closer to a ~100 ky old Siberian
wolf or later individuals.

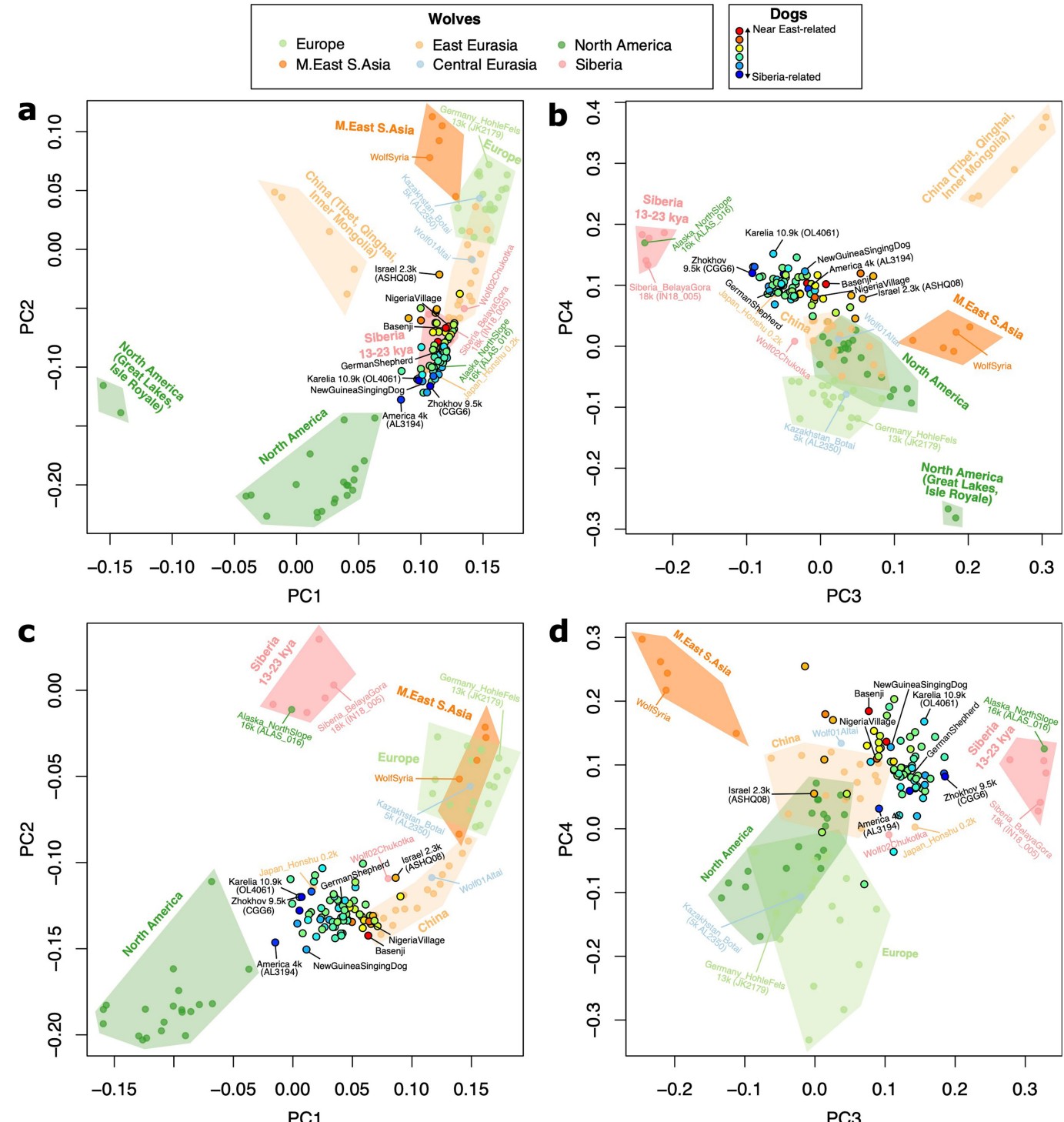

**Extended Data Fig. 2 | Placing dogs into wolf diversity in a 'pre-LGM $f_4$ PCA'.** PCA on wolves that lived after 25 ka (including present-day), based on profiles of $f_4$-statistics only of the form $f_4(X,A;B,C)$, where A, B, and C are wolves that lived prior to 28 kya. Dogs are projected. Dogs are coloured according to the $f_4$-statistic $f_4(AndeanFox,X;Zhokhov\ dog\ 9.5ka,Tel\ Hreiz\ dog\ 7.2ka)$, with negative values going towards blue and positive values towards red. A few wolves

(in colour) and dogs (in black) of particular interests are indicated with text labels. a) PC1 vs PC2 with the full set of wolves. b) PC3 vs PC4 with the full set of wolves. c) PC1 vs PC2 with western Chinese and North American outlier wolves removed. d) PC3 vs PC4 with western Chinese and North American outlier wolves removed.

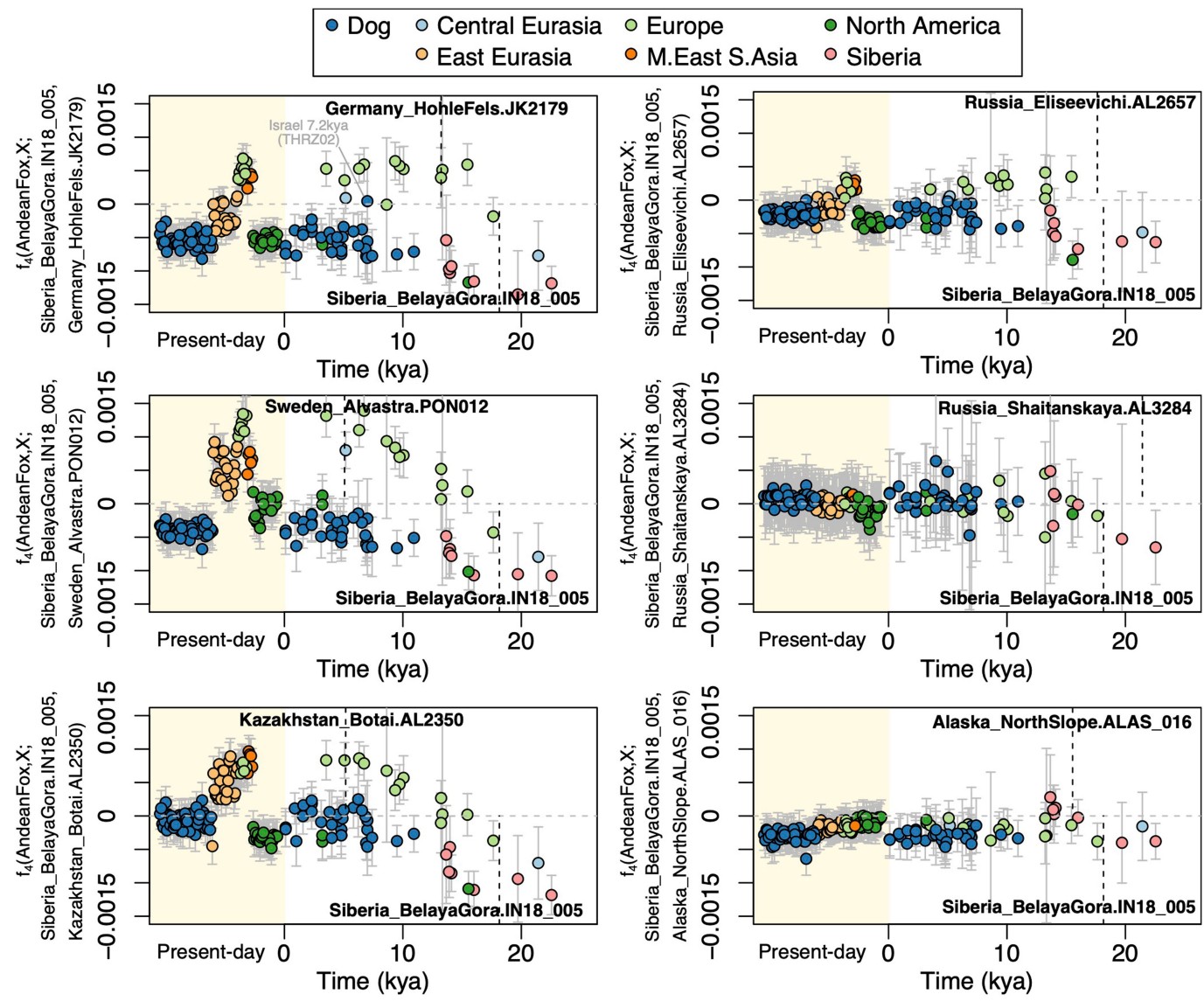

**Extended Data Fig. 3 | Affinities of dogs to ancient wolves.** a) $f_4$-statistics of the form $f_4(AndeanFox,X;wolf A,wolf B)$, quantifying for all individuals X whether they share more drift with wolf A or wolf B. The ages of A and B are indicated with dashed lines, with positive values indicating affinity to the upper individual and negative values indicating affinity to the lower individual. Bars denote ±3 standard errors estimated from a block jackknife.

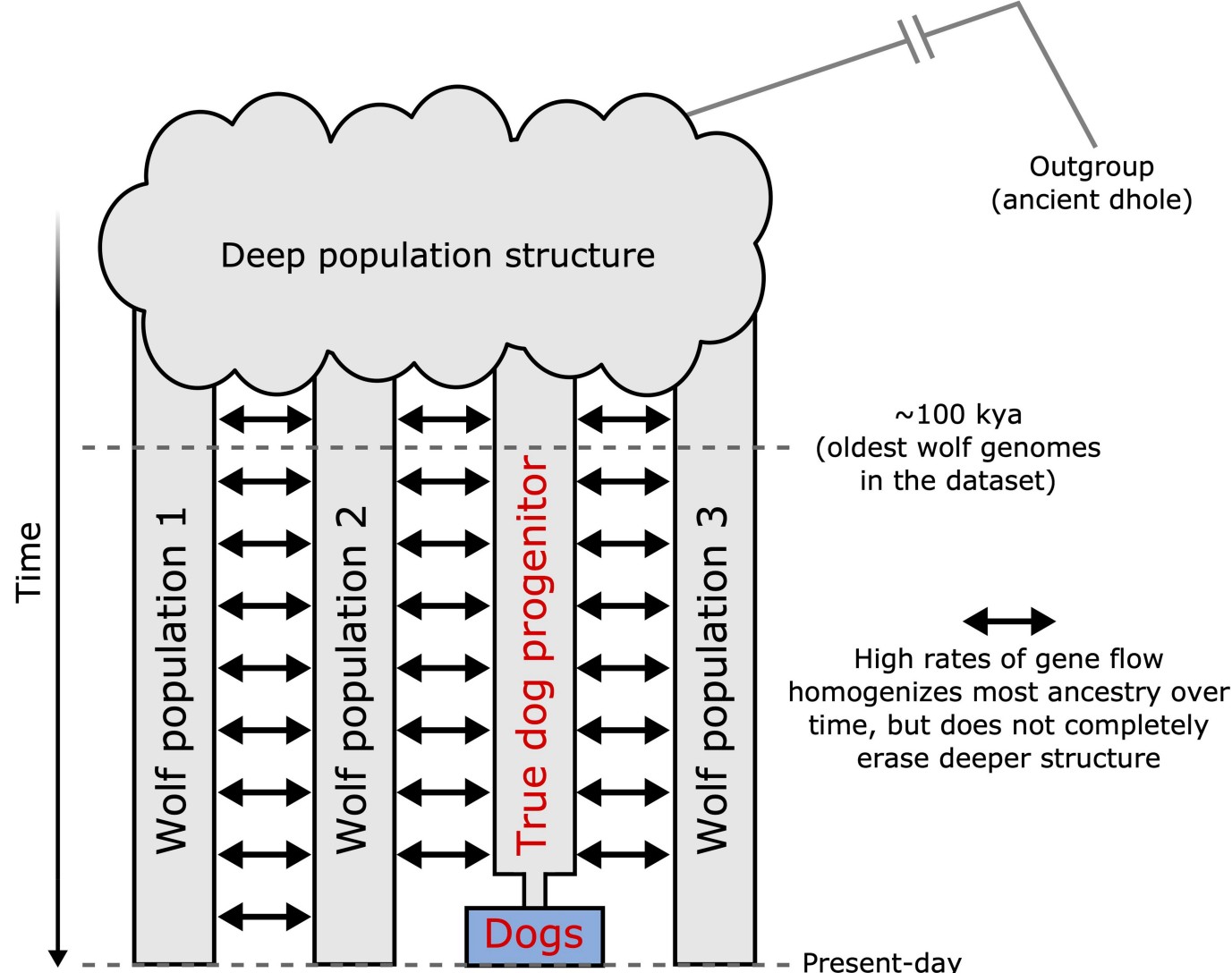

**Extended Data Fig. 4 | A schematic model of how deep population structure could explain why dogs require ancestry from an outgroup population in *qpAdm* analyses.** Under this model, there is deep population structure between different wolf populations, including the wolf population that becomes the progenitor of dogs. High rates of gene flow over time largely homogenises the ancestry of all populations, but it does not completely erase the deep structure. If the true dog progenitor population is not sampled, a single-source qpAdm model involving one of the sampled wolf populations will not fit dog ancestry, because dogs do not share all of the genetic drift that has occurred in the history of the sampled population. But if an outgroup population is included as a source in qpAdm, this can account for the 'missing' deep ancestry in dogs, and therefore result in a model that fits dog ancestry.

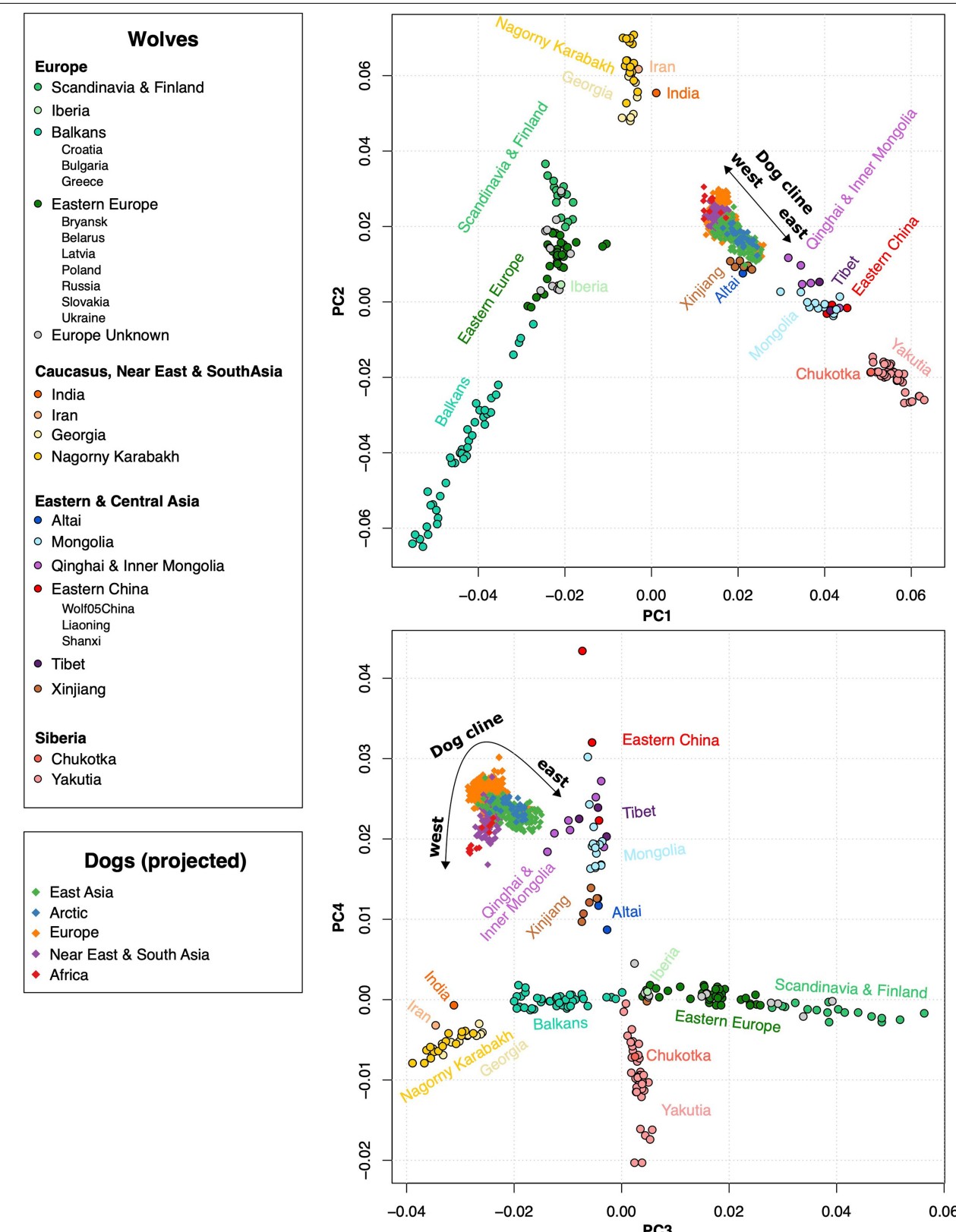

**Extended Data Fig. 5 | Projecting dogs onto present-day wolf population structure.** Principal components analyses performed only on modern wolves, with modern dogs projected.

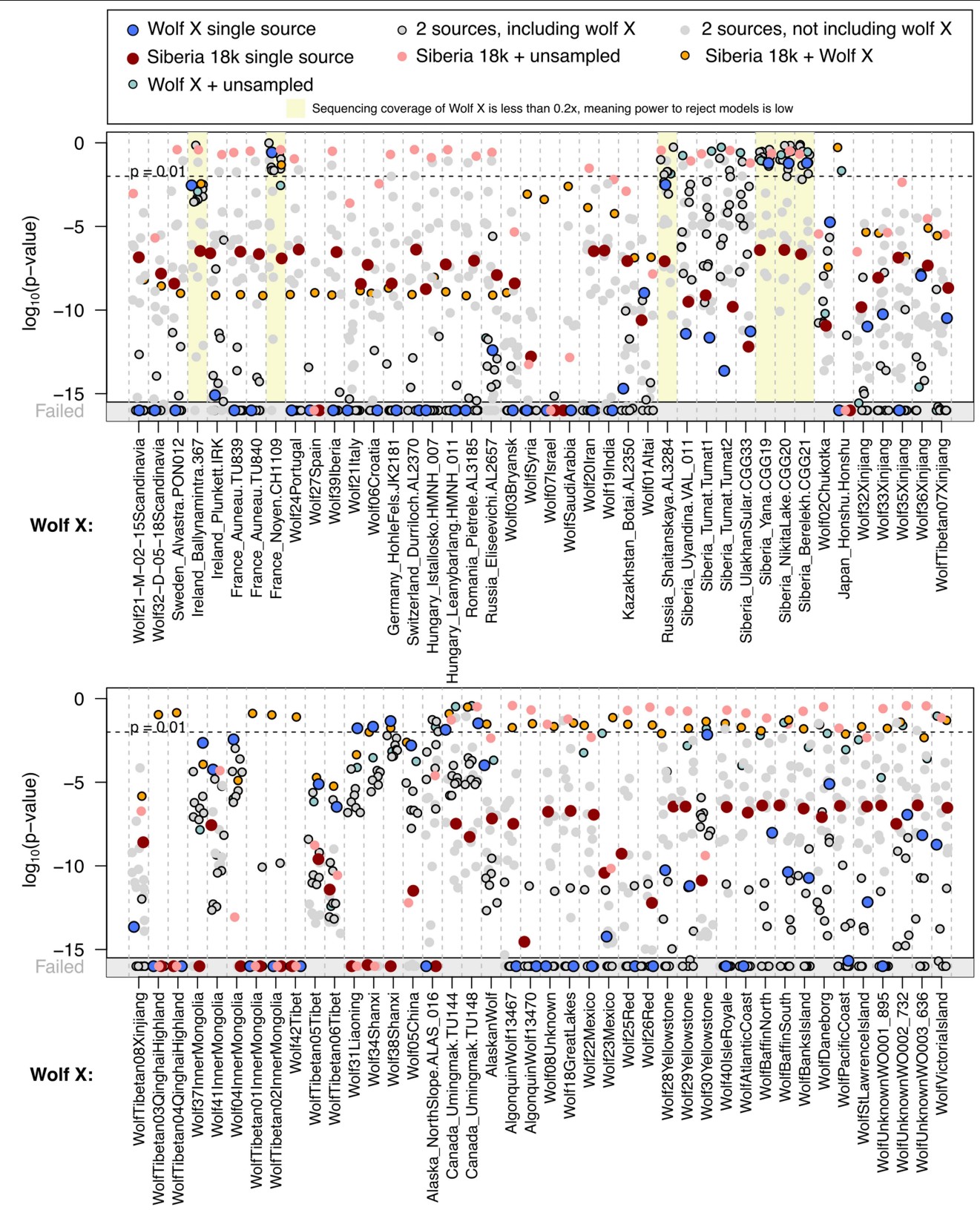

**Extended Data Fig. 6 | "Ocean plot" searching for the best available wolf match for the ancestry of eastern dogs.** With the Siberian Zhokhov dog (9.5k BP) as the target, each candidate wolf X was added in turn into the rotating qpAdm analysis. When X is not part of the sources, it is placed in the reference list. Models placed within the gray space labelled "Failed" have p-values fall below the lower limit of the plot.

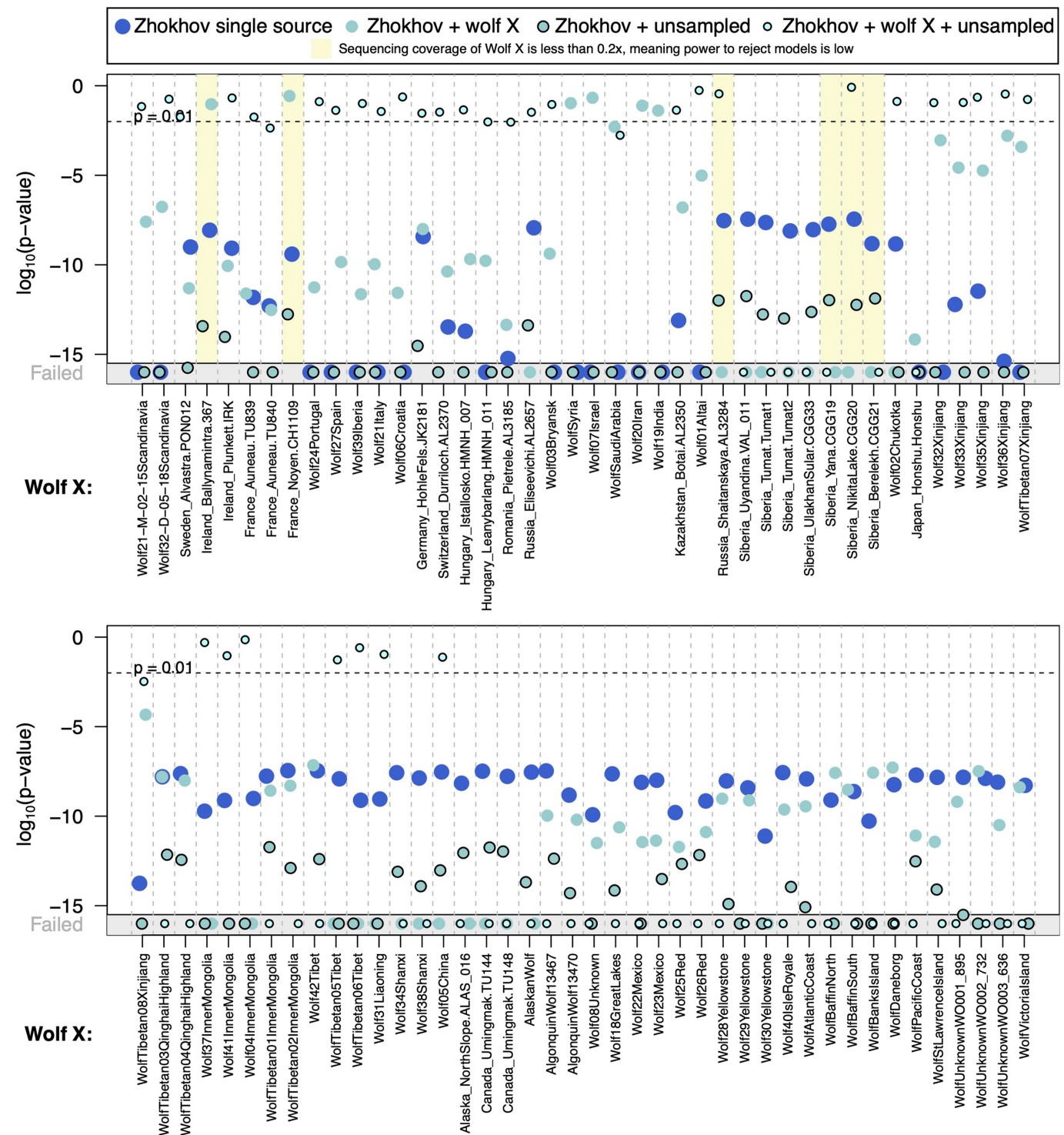

**Extended Data Fig. 7 | "Ocean plot" searching for the best available wolf match for the west Eurasian wolf-related ancestry in western dogs.** With the African Basenji dog as a target, all available post-LGM and present-day wolf genomes X are tested as sources combined with the 9.5k-year old Siberian Zhokhov dog, which is assumed to represent a baseline for the Eastern-related dog progenitor ancestry. When X is not part of the sources, it is placed in the reference list. If a target has a model with p > 0.01, models with a larger number of sources are not plotted. Only four individuals achieve good fits in the two-source model (Zhokhov + X): WolfSyria, Wolf07Israel, Wolf20Iran and Wolf19India. For other individuals, including ancient and present-day European wolves, the two-source model can be rejected, and a three-source model with an unsampled ancestry component (Zhokhov + X + unsampled) is needed to fit the data.

**Extended Data Table 1 | Selection peaks**

| Chr | Start (Mb) | End (Mb) | Description and notes on genes within region |
|---|---|---|---|
| 1 | 103.7 | 103.8 | *ZNF331*, zinc-finger protein involved in transcriptional regulation |
| 2 | 6.77 | 6.84 | *YME1L1* involved in mitchondrial morphology, highly expressed in muscle. Mutations in humans associated with optic atrophy |
| 3 | 72.35 | 72.45 | *N4BP2* may play a role in DNA repair or recombination |
| 4 | 32.22 | 32.25 | No genes |
| 6 | 9.85 | 9.95 | *CYP3A26* is a cytochrome P450 enzyme |
| 6 | 13.85 | 14.05 | No genes |
| 6 | 43.8 | 43.82 | *VAV3* involved in angiogenesis |
| 7 | 29 | 29.05 | *F5* is coagulation factor V, *SELL* has immunity function |
| 9 | 2.2 | 2.3 | No genes, lncRNA |
| 9 | 8.95 | 9.6 | *KANSL1* associated to Koolen-de Vries hypersociability syndrome, *MYL4* is involved in muscle function |
| 10 | 7.62 | 7.7 | Dog QTL locus associated to drop ears, body mass and other traits. *WIF1* inhibits Wnt signalling, role in embryonic development |
| 10 | 7.95 | 8.09 | Dog QTL locus associated to drop ears, body mass and other traits. Human mutations in *MSRB3* associated to deafness |
| 10 | 8.14 | 8.24 | Dog QTL locus associated to drop ears, body mass and other traits. |
| 11 | 0.75 | 1.15 | *OR2AI2* is olfactory receptor, *IFGGB2* has immunity function |
| 11 | 56.72 | 56.77 | No genes |
| 15 | 0.1 | 0.5 | Olfactory gene cluster, SLC2A1 is Glucose transporter 1 |
| 15 | 3.92 | 3.98 | No genes |
| 15 | 6.53 | 6.57 | *TFAP2E* linked to Branchiooculofacial Syndrome which includes facial development problems |
| 15 | 13.5 | 13.7 | Three cytochrome P450 enzyme genes, involved in lipid and secondary metabolism |
| 21 | 28.02 | 28.07 | Olfactory gene cluster |
| 22 | 2.8 | 2.92 | lncRNA, just downstream of *CYSLTR2* |
| 25 | 17.4 | 17.56 | *IFT88*, involved in craniofacial development |
| 25 | 19.77 | 19.9 | Uncharacterized gene |
| 30 | 2.69 | 2.75 | No genes |

Locations in the genome of regions displaying evidence of natural selection across the wolf time series, with comments on any genes within the region. For a more detailed table see Supplementary Data 3.

## Extended Data Table 2 | qpWave tests of dog cladality

| Target sets | Individuals |
|---|---|
| Eastern dogs | Karelia_Veretye.OL4061, Zhokhov.CGG6, PortauChoix.AL3194, Baikal.OL4223, NewGuineaSingingDog |
| Southwestern dogs | Israel.THRZ02, Iran.AL2571, Israel.ASHQ01, Basenji |

| Ancient reference sets | Individuals |
|---|---|
| Ancient small (n=7) | Siberia_UlakhanSular.LOW008, Germany_Aufhausener.AH575, Germany_HohleFels.JK2183, Siberia_BungeToll.CGG29, Siberia_BelayaGora.IN18_016, Yukon_QuartzCreek.SC19.MCJ010, Altai_Razboinichya.AL2744 |
| Ancient large (n=25) | Germany_Aufhausener.AH574, Germany_Aufhausener.AH577, Siberia_Yana.CGG27, Siberia_Badyarikha.CGG34, Alaska_Fairbanks.JAL385, Alaska_Fairbanks.JAL48, Alaska_Fairbanks.JAL65, Alaska_Fairbanks.JAL69, Yukon_HunkerCreek.SC19.MCJ017, Germany_HohleFels.JK2174, Germany_HohleFels.JK2175, Germany_HohleFels.JK2183, Siberia_BungeToll.LOW003, Siberia_UlakhanSular.LOW008, Czechia_Predmosti.PDM100, Alaska_LillianCreek.ALAS_024, Siberia_Tirekhtyakh.VAL_033, Siberia_Badyarikha.VAL_008, Siberia_Ogorokha.VAL_050, Siberia_BelayaGora.IN18_016, Siberia_Tirekhtyakh.CGG32 |

| Target | Reference set | p rank 0 | p rank 1 | p rank 2 |
|---|---|---|---|---|
| Eastern dogs | Ancient small | 0.3667 | 0.9566 | 0.9992 |
| Southwestern dogs | Ancient small | 0.0229 | 0.8850 | 0.8474 |
| Eastern+Southwestern | Ancient small | 6.1E-05 | 0.1900 | 0.7610 |
| Eastern dogs | Ancient large | 0.0656 | 0.5352 | 0.8292 |
| Southwestern dogs | Ancient large | 0.1622 | 0.8989 | 0.9525 |
| Eastern+Southwestern | Ancient large | 9.2E-18 | 2.9E-04 | 0.0659 |

| Modern reference sets | Individuals |
|---|---|
| Base modern | WolfSaudiArabia, WolfSyria, Wolf01Altai, Wolf02Chukotka, Wolf03Bryansk, Wolf04InnerMongolia, Wolf05China, Wolf06Croatia, Wolf07Israel, Wolf19India, Wolf20Iran, Wolf21Italy, Wolf24Portugal, Wolf27Spain, Wolf31Liaoning, Wolf32Xinjiang, Wolf33Xinjiang, Wolf34Shanxi, Wolf35Xinjiang, Wolf36Xinjiang, Wolf37InnerMongolia, Wolf38Shanxi, Wolf39Iberia, Wolf41InnerMongolia, Wolf42Tibet, WolfTibetan01InnerMongolia, WolfTibetan02InnerMongolia, WolfTibetan03QinghaiHighland, WolfTibetan04QinghaiHighland, WolfTibetan05Tibet, WolfTibetan06Tibet, WolfTibetan07Xinjiang, WolfTibetan08Xinjiang, Wolf21-M-02-15Scandinavia, Wolf32-D-05-18Scandinavia |

| Target | Reference set | p rank 0 (mean log across reps) | p rank 0: max | p rank 1 (mean log across reps) | p rank 1: max |
|---|---|---|---|---|---|
| Eastern dogs | (Sample of n=7 from base) x100 reps | 2.5E-11 | 0.0214 | 0.2643 | 0.9576 |
| Southwestern dogs | (Sample of n=7 from base) x100 reps | 0.0474 | 0.9247 | 0.3226 | 0.9990 |
| Eastern+Southwestern | (Sample of n=7 from base) x100 reps | 6.7E-77 | 1.2E-11 | 3.0E-05 | 0.4611 |
| Eastern dogs | (Sample of n=25 from base) x100 reps | 2.2E-46 | 1.1E-23 | 0.0033 | 0.3526 |
| Southwestern dogs | (Sample of n=25 from base) x100 reps | 7.1E-06 | 0.0525 | 0.0427 | 0.8939 |
| Eastern+Southwestern | (Sample of n=25 from base) x100 reps | 1.0E-100 | 1.0E-100 | 1.2E-54 | 1.0E-06 |

Two different dog target sets, and their union, are tested for cladality relative to reference sets consisting of ancient or modern wolves. From the modern wolves (bottom of table), for each target 100 different reference sets were constructed by randomly sampling either 7 or 25 individuals. The results across these 100 tests are summarised by displaying the mean (on a log-scale) and maximum p-values.

# Reporting Summary

## Statistics

For all statistical analyses, confirm that the following items are present in the figure legend, table legend, main text, or Methods section.

| n/a | Confirmed | |
|---|---|---|
| ☒ | ☐ | The exact sample size (*n*) for each experimental group/condition, given as a discrete number and unit of measurement |
| ☒ | ☐ | A statement on whether measurements were taken from distinct samples or whether the same sample was measured repeatedly |
| ☐ | ☒ | The statistical test(s) used AND whether they are one- or two-sided *Only common tests should be described solely by name; describe more complex techniques in the Methods section.* |
| ☐ | ☒ | A description of all covariates tested |
| ☐ | ☒ | A description of any assumptions or corrections, such as tests of normality and adjustment for multiple comparisons |
| ☐ | ☒ | A full description of the statistical parameters including central tendency (e.g. means) or other basic estimates (e.g. regression coefficient) AND variation (e.g. standard deviation) or associated estimates of uncertainty (e.g. confidence intervals) |
| ☐ | ☒ | For null hypothesis testing, the test statistic (e.g. *F*, *t*, *r*) with confidence intervals, effect sizes, degrees of freedom and *P* value noted *Give P values as exact values whenever suitable.* |
| ☐ | ☒ | For Bayesian analysis, information on the choice of priors and Markov chain Monte Carlo settings |
| ☒ | ☐ | For hierarchical and complex designs, identification of the appropriate level for tests and full reporting of outcomes |
| ☐ | ☒ | Estimates of effect sizes (e.g. Cohen's *d*, Pearson's *r*), indicating how they were calculated |

*Our web collection on statistics for biologists contains articles on many of the points above.*

## Software and code

Policy information about availability of computer code

| Data collection | No software was used for data collection. |
|---|---|
| Data analysis | SeqPrep, BWA aln v.0.7.17, BWA mem v0.7.15, PMDtools v0.60, Picard Tools v2.21.4, GATK HaplotypeCaller v3.6, bcftools v.1.8, htsbox pileup r345, samtools v1.9, Geneious v9.0.5, Clustal Omega v1.2.4, BEAST v1.10.1, JModelTest2 v2.1.10, ADMIXTOOLS v5.0, R package pcaMethods v1.74.0, EIGENSOFT 7.2.1, gem-mappability v1.315, MSMC2 v2.1.2, PLINK v1.90b5.2, ms, SHAPEIT4 v4.2.1, Relate v1.1.8, CLUES, Integrative Genomics Viewer (IGV) 2.4.14, R package mapdata 2.3.0, OxCal v4.4 |

For manuscripts utilizing custom algorithms or software that are central to the research but not yet described in published literature, software must be made available to editors and reviewers. We strongly encourage code deposition in a community repository (e.g. GitHub). See the Nature Portfolio guidelines for submitting code & software for further information.

## Data

Policy information about availability of data

All manuscripts must include a data availability statement. This statement should provide the following information, where applicable:

- Accession codes, unique identifiers, or web links for publicly available datasets
- A description of any restrictions on data availability
- For clinical datasets or third party data, please ensure that the statement adheres to our policy

The generated DNA sequencing data is available in the European Nucleotide Archive (ENA) under study accession PRJEB42199. Previously published genomic data analysed here is available under accession numbers PRJNA448733, PRJCA000335, PRJEB20635, PRJNA496590, PRJNA494815, PRJEB7788, PRJEB13070, PRJNA319283, PRJEB22026, PRJNA608847, PRJEB38079, PRJEB39580 and PRJEB41490, with individual genomes used listed in Supplementary Data 2. The

canFam3.1 reference genome is available under NCBI assembly accession GCF_000002285.3.

# Field-specific reporting

Please select the one below that is the best fit for your research. If you are not sure, read the appropriate sections before making your selection.

☒ Life sciences    ☐ Behavioural & social sciences    ☐ Ecological, evolutionary & environmental sciences

For a reference copy of the document with all sections, see nature.com/documents/nr-reporting-summary-flat.pdf

# Life sciences study design

All studies must disclose on these points even when the disclosure is negative.

| | |
|---|---|
| Sample size | No sample size calculations were made. This is a genomic study of paleontological material, where sample size was shaped by availability of material (ancient wolf remains) and their DNA preservation upon screening. This is the largest ancient genomic study of Pleistocene genomes to date. The sample size of over 70 ancient wolves and hundreds of canids in the published literature provides substantial statistical power, notably due to the evolutionary variance being accounted for by analysis of the entire genome - comprising tens of hundreds of thousands of independent loci. |
| Data exclusions | All genome sequencing data collected for this study was analyzed. Certain population genetic analyses were restricted to subsets of genomes (e.g. those meeting thresholds of sequencing coverage or other measures of data quality) as detailed in the Methods section and Supplementary Information. |
| Replication | This was a retrospective study of an evolutionary history that has occurred only once, and it was not possible to observe independent replicates of this history. |
| Randomization | This was a retrospective study of an evolutionary history that has occurred only once, and it was not possible to randomize the application of different past processes to the analyzed genome sequences. |
| Blinding | Blinding was not applicable to this study, as each genome sequence had to be associated with its spatial and temporal metadata in order to draw conclusions. |

# Reporting for specific materials, systems and methods

We require information from authors about some types of materials, experimental systems and methods used in many studies. Here, indicate whether each material, system or method listed is relevant to your study. If you are not sure if a list item applies to your research, read the appropriate section before selecting a response.

### Materials & experimental systems

| n/a | Involved in the study |
|---|---|
| ☒ | ☐ Antibodies |
| ☒ | ☐ Eukaryotic cell lines |
| ☐ | ☒ Palaeontology and archaeology |
| ☒ | ☐ Animals and other organisms |
| ☒ | ☐ Human research participants |
| ☒ | ☐ Clinical data |
| ☒ | ☐ Dual use research of concern |

### Methods

| n/a | Involved in the study |
|---|---|
| ☒ | ☐ ChIP-seq |
| ☒ | ☐ Flow cytometry |
| ☒ | ☐ MRI-based neuroimaging |

## Palaeontology and Archaeology

| | |
|---|---|
| Specimen provenance | The metadata for the 67 wolf remains from which novel genome sequencing data is reported is described in the table in Supplementary Data 1. For each specimen, this table lists the name and geographical coordinates of the site of excavation or collection, the steward institution that provided access to and is responsible for the long-term storage of the specimen, the excavation or museum collection identifier if applicable, and what skeletal element was sampled for the purpose of DNA extraction. As no new excavations were performed in this study, no excavation permits were necessary. Sampling for DNA extraction was performed with the permission of the specimen stewards, all of which are listed in Supplementary Data 1, and most of which are authors on the paper. |
| Specimen deposition | The metadata table in Supplementary Data 1 lists, for each of the 67 wolf remains from which novel genome sequencing data is reported, the steward institution that provided access to and is responsible for the long-term storage of the specimen, and the excavation or museum collection identifier if applicable. Requests for access to the specimens should be directed to these host institutions. |

Dating methods | New radiocarbon dates were obtained from the Oxford Radiocarbon Accelerator Unit and calibrated using the IntCal20 calibration curve in the OxCal v4.4 software. We refer to the dating laboratory for details on their experimental protocol.

☒ Tick this box to confirm that the raw and calibrated dates are available in the paper or in Supplementary Information.

Ethics oversight | No ethical oversight was required as this study comprises only zooarchaeological material, previously collected and curated by individual institutions and researchers following local regulations. Sampling for DNA was performed aiming to minimize the destructive impact on the zooarchaeological material.

Note that full information on the approval of the study protocol must also be provided in the manuscript.

