## [Peer Review File · Nature]

Manuscript Title: Gray wolf genomic history reveals a dual ancestry of dogs

Reviewer Comments & Author Rebuttals

Reviewer Reports on the Initial Version:

Referees' comments:

Referee #1 (Remarks to the Author):

The manuscript "Grey wolf ancestry reveals a dual history of dogs" by Bergstrom et al analyzes a comprehensive repertoire of ancient DNA samples from the wolf and dog lineages to assess the evolutionary history of both species. The publication applies evolutionary analysis of genetic drift via f_3 and f_4 statistics. These show a temporal pattern of genetic drift suggesting regular admixture among populations in the last 100k years. In more recent times, the Siberian wolf population diverges to affect domestic dog ancestry in Europe. In North America, admixture with Coyote obfuscates simple analysis.

The study expands on (and somewhat contradicts) previous work by the same authors who previously suggested that dogs came from a single common wolf lineage (<https://www.science.org/doi/10.1126/science.aba9572>). The results are of interest to canine biologists and evolutionary biologists. The close affinity between domestic dogs and humans through history means that others may also be interested.

Data & methodology: There are a few aspects of the work that might be improved.

The manuscript is well analyzed applying appropriate methods and strategies to form conclusions from the data. While the text reads well, and the figures have attractive colors, more care could be taken to help the non-evolutionary biologist understand what is depicted in the figures. Most of the data provided use appropriate statistics and error bars but the results - particularly in figures could be better explained (see comments following).

Conclusions: The publication makes use of expanded wolf ancestral data to inform canid evolutionary history and in particular the history of wolves. There remain some things to be explained, as at least one basal ancestor is yet to be determined in the future. Determining this ancestor is not possible with current resources.

References: Referencing is high quality.

Suggested improvements:

First, the charts need to be reorganised to enable a single legend for multi-part panels (e.g. Extended Figure 1 a and b). The flipping between a y-axis of $-\log(P)$ and $\log(P)$ on chart axes is confusing to

follow. In general, there is insufficient information in the Figure descriptions to understand what they depict. The numbering of supplementary figures in the text is all over the place and Figure S1 is not referred to at all.

Fig 1b legend should mention which samples (presumably wolves and dhole as no dog - but I do not see the dhole). Figure 1c, it is unclear which samples are represented in this chart -maybe give a number and refer to a supp table for description?

The explanation for Fig 4c should be clarified to help the novice reader interpret the chart (and is incorrect as it lists that the p-value is plotted rather than the log (P-value)) and should be clear that strong significance refutes rather than supports the proposed ancestry. In Fig 4c the markers do not always seem to align properly in the columns for each dog. Is it intentional that North America is absent from the world map in Fig 4e? As a block the Figure 4 does not "stand alone".

Figure S1 is not referenced in the main text.

Figure S9 description "A positive value of this statistic implies that the non-canid species is closer to X, and a positive value implies it's closer to Siberia_Enygen.VAL_18A."

Is this correct? Should one say negative?

Main paper lines 153-4 - I am not certain from Figure S9 how this conclusion was reached?

Lines 162-3 Doesn't Figure S9 (first panel) strongly suggest Coyote common ancestry?

Figure S8 -please tell us explicitly that the up and down arrows in the titles relate to x and y axes respectively?

The description of "basal" ancestry, provided on lines 862-864 should be provided at least briefly earlier (e.g. before Line 87).

Clarity and context: The abstract provided is clear and outlines the major findings from the work.

Minor comments:

Fig 1b (not bold) line 119.

Referee #2 (Remarks to the Author):

Bergstrom et al. report findings of an immense undertaking, by way of sequencing 67 new ancient wolf genomes and an ancient dhole, and jointly analyzing these with previously published ancient and modern dog and wolf genomes. Their objectives are twofold: to more deeply understand the demographic history of wolves, and to hopefully use this information to gain new insight into the origin of dogs. The very much correct premise of this work is that analysis of ancient wolf genomes is crucial to understanding dog domestication, as more recent patterns of population expansion/contraction and admixture—both between wolf populations and between wolves and dogs—will obscure dog origins when one largely works with contemporary samples. The author's major findings are that populations structure in wolves during the Pleistocene was weaker than it is

today; the majority of wolf ancestry in contemporary populations (including Europe and North America) originates from Siberia, but variable fractions of deep local ancestry persist outside of eastern Eurasia; the majority of dog ancestry originates from eastern Eurasia, but none of the populations represented by the ancient Siberian wolf samples appears to have been the direct ancestral source of dogs; and finally, that a minority fraction of dog ancestry originates from a western Eurasian source, either through admixture with local wolves or as the result of a second domestication process.

Broadly speaking, the presented work represents a significant advance in our understanding of wolf population history and dog domestication, ruling out a northern European origin for dogs and getting us geographically closer to the source population(s) from which dogs originated. But, in the absence of any definitive resolution regarding the geographic origin of dogs, I am on the fence as to whether the presented work represents a major advance in our understanding as to the origin of dogs.

Overall, the paper is clearly written and I have no major concerns with methods or inferences drawn from them, and what concerns/questions I have follow below.

1. The selection analyses are the weakest part of the paper and detract from the story line around the main points. They are based upon single-SNP analyses—even if those SNPs are detected based upon a minimum signal in adjacent SNPs. In 2021, without functional tests or phenotypic associations, my understanding is that top tier journals such as Nature are disinclined to publish such results as they smack of storytelling. Too many studies have produced different selection hits solely by employing different methods. My opinion is that the selection results should be largely relegated to the supplementary information. I'd rather see Extended Data Figure 1 (or at least parts of it) moved to the main figures, replacing Figure 3, as it is central to the new findings on wolf demographic history.
2. Figure 1a. It took me a minute to orient myself geographically since the figure is essentially looking down on the north pole. Perhaps some latitudinal lines would make this clearer or some other means of geographical orientation on the figure?
3. In the paper there is reference to a continuous process of ancestry turnover but looking at Extended Data Figure 1b, and Figure S4—there does seem to be a shift over 10k years to largely ancestry originating from ancient Siberian wolf samples. This ancestry is a major point of the paper, and this not very continuous change is biologically interesting, so perhaps a shift in language is warranted along with at least some speculation as to what is driving what looks like a step function in the F4 statistics.
4. Lines 252-256 regarding timing of dog domestication. It is unclear how the statements wrt the separation of dogs and wolves are supported by either Fig. 1c or Extended Data Fig 1b. Also, I think the authors meant to say “We found that dogs were not separated from wolves until at least 28kya as they share ancestry with all wolves predating this time...” whereas they said “postdating.” I think there are other places in the manuscript where the authors language is unclear wrt whether they are looking forward or backward in time. That said, the shared drift plots can only make inferences on dog “separation” from wolves based upon the dates of the ancient dog samples, all of which postdate most genetic estimates of dog origins. Regarding the estimation of dog-wolf divergence from MSMC2 plots based upon artificially constructed X diploids, there are two potential issues.

First, the rapid increase in N_e deemed to be a signature of divergence could also be produced by other factors, namely admixture with another lineage, even if that admixture affected both populations from which the X chromosomes were drawn. It is easy to imagine a case where a) admixture with a third divergent lineage produces elevated heterozygosity and 2) the sampling of one X per population leads to each X including alternative alleles at the heterozygous sites produced by admixture. There truly is elevated heterozygosity in both target populations being analyzed but it has nothing to do with divergence. The strength of such an effect would be influenced by the population minor allele frequencies (at heterozygous sites) and the timing of admixture events. Second, it is challenging enough to hypothesize divergence events from MSMC plots comprised of true diploid individuals, let alone pseudohaploids comprised of low coverage genomes. Third, in the Supplementary Material, several of the MSMC plots indicate even older hypothesized dates of dog-wolf divergence. Overall, I think strong caveats need to be placed on the authors' claim of a more recent dog-wolf divergence.

5. Wrt the inferred second source of dog ancestry, I feel more confident in the results obtained from modeling ancestry contributions via qpWave, qpAdm and qpGraph that I do with the PCA results presented in Fig 4a. The effect in the PCA plot by way of the shift of some dogs towards the SW Asia is in most cases subtle, and the more dramatic cases being few. The crowding of points, and the inability to know which points represent which breeds makes interpretation even harder: perhaps a blow up of that part of PCA space where the dogs are and some breed labels would be helpful in this regard. Furthermore, one can't rule out that some of these shifts in PCA space aren't due to recent, geographically specific admixture between dogs and wolves as has been reported previously in other studies. If the ancestry modeling analyses were framed in the text as a way to overcome the potential confounding of signals in the PCA that would largely ameliorate the issue.

Referee #3 (Remarks to the Author):

Overview

Bergström and colleagues present nuclear genomic data from a truly impressive number of ancient gray wolf specimens. This is now the most densely sampled temporal and geographical genomic transect for any non-human animal species - it beats the ancient sampling for both mammoths and horses. The framework it provides is compelling: by characterising the pre-domestication distribution of wolf diversity, the authors can more precisely identify the timing, location, and number of dog domestication events. Bergström and colleagues rigorously test a huge range of ancestry scenarios and are able to confidently reject a number of hypotheses, which greatly narrows the range of plausible origins for modern domestic dogs (and modern wolves too). The authors central claim - a dual ancestry for modern dogs - appears to be beyond reasonable doubt, as the most parsimonious explanation for the observed ancestry asymmetries observed in modern wolves and dogs. This study also outlines clear priorities for follow-up research, namely sampling of additional ancient wolves from central Eurasia and the contiguous US. I congratulate the authors on an outstanding piece of work.

I have no major issues with the authors' methodology or their interpretation of the results, with the

exception of some reservations about the strength with which the authors assert that fixation of several alleles was caused by selection as opposed to neutral processes (see below). The rest of my feedback mainly centres on clarity of communication and accuracy of terminology. With corresponding revisions - which I have no doubt the authors can satisfactorily perform - I would be pleased to recommend this excellent study for publication.

General comments

I really like Extended Figure 3b (the plot of f_4 results for modern dogs against two ancient wolves versus the f_4 results for the same modern dogs against two ancient dogs) and I felt it was one of the clearest and most compelling illustrations of the whole manuscript's main message. I would strongly encourage the authors to move it to the main text. In my opinion it could easily replace Fig. 4a - I found the PCA difficult to interpret anyway because the dogs are coloured by their ancestry and not geographical origin. I have to admit I also didn't find Fig. 4b to be strictly necessary; since it's only text anyway the same information could easily just be summarised in the paragraph on Lines 304-311.

The fixation of alleles in both IFT88 & the olfactory gene cluster on chr15 appears to have occurred post 28 kya (Lines 215-243, Fig. 3b). But in the paragraph immediately prior (Lines 202-212) the authors flag this time period as the onset of wolf population structure, resulting from some change in demographic processes (either decline in effective population size, reduction in gene flow, or possibly both). As a result, this is the exact time period when drift would be expected to disproportionately result in false positive signatures of selection using the method the authors employed (which does not correct for demography). They are certainly outliers and still interesting signals - I don't debate that - but I'm not personally convinced on the basis of the results presented that they must necessarily result from selection (versus simply neutral processes). I'd be more comfortable if the authors acknowledged or addressed this caveat in the main text. Overall, when compared to the other sections of this study - where the data are explored and scrutinised exhaustively with multiple analyses - this section on selection stood out to me as relatively bare-bones. So the authors might also consider omitting this section - thereby strengthening the central narrative about ancestry and origins - and instead follow up with a more in-depth study focused on adaptation (which I suspect they may already be planning anyway).

Specific comments

Line 150 "...ancestry that is basal to the oldest Siberian wolves..."

TL;DR - Please revise usage of "basal" throughout.

The authors use "basal" here and throughout the manuscript, but the term is not defined for the reader until the end of the methods on Lines 864-865: "basal to (i.e. diverging early from, and lacking shared genetic drift with)". Even then, this definition differs from the meaning and usage of the word in phylogenetics, and risks perpetuating a persistent misconception (see summary by Krell & Cranston 2004; <https://doi.org/10.1111/j.0307-6970.2004.00262.x>). In brief, "basal" simply and strictly refers to a part of the tree that is closer to the root than another (i.e. older). Usually the term

applies to nodes, but it can also apply to taxa/lineages/populations (hereafter just taxa) but - importantly! - only when the representatives of those taxa are not coeval (e.g. a 20 kya wolf is more basal than a modern wolf, but two 20 kya wolves are equally basal). Therein lies the problem with the usage of basal in this paper, because introgressed ancestry does not derive from nodes, it derives from taxa. And admixture can only occur between coeval taxa. Because - as above - neither coeval taxon can be regarded as basal with respect to another (they are equally old, equally distant from their common ancestor), the ancestry inherited from both parties is equally basal. So - with the possible exception of time travel shenanigans - the concept of "basal ancestry" doesn't really make sense.

What the authors are actually referring to is ancestry from an unsampled or "ghost" lineage. They already specifically invoke this concept on Line 285-286 in reference to the ancestry of the Zhokov dog: "We therefore interpret the results for dogs as similarly reflecting some unsampled wolf ancestry..." But the "basal ancestry" referred to on Line 150 is also represented as "unsampled wolf ancestry" in Figure 1C, though - importantly - in these two cases the ancestry is coming from two different unsampled lineages: the former ancestry donor is a member of a sister lineage to ancient Siberian wolves, and the latter a sister lineage to all sampled modern and ancient wolves. It would be more accurate - and much clearer for the reader - if the authors entirely abandoned the use of "basal" and instead explicitly described the affinities of the unsampled lineage in each case. For example, the sentence beginning on Line 148 could instead read: "Most analysed present-day Eurasian wolves likely retain local Pleistocene ancestry, as they are best modelled by qpAdm as having 10-40% ancestry from an unsampled sister lineage to all modern and ancient wolves analysed in this study (Fig. S10, S11)." Where "basal ancestry" is used in the key for figures, it could easily be replaced by "unsampled ancestry", "ghost ancestry", or similar as appropriate.

Line 165-166 "...two Pleistocene Alaskan wolves also carry mitochondrial lineages that have been found in coyote"

There are six Pleistocene samples from Alaska in the dataset. Which two have the coyote-like mitochondrial haplotypes? And as a follow-up, are the the two ancient wolves with coyote haplotypes mentioned here the same individuals as the two non-modern coyotes mentioned on Line 757: "...included 3 coyotes as outgroups (1 modern, 1 14C dated and 1 with an infinite 14C date)"? If so, its not strictly correct to refer to them as coyotes in the second instance.

This might seem a trivial thing, but the ages for three of those six Alaskan wolf samples are inferred based only on the mtDNA analyses. The minimal representation of coyote mitochondria in the dataset could mean that the estimated ages for the two ancient samples with coyote-like haplotypes are biased (or at least imprecise). It's unlikely to be seriously problematic, but it would be useful to flag these two samples - perhaps in the sample metadata table - so that readers can easily cross-reference and judge for themselves.

I also found the intent of the in-text citation on Line 166 a little ambiguous. I originally read it as meaning that Vila et al. 1999 had already identified these two mitochondrial haplotypes in Pleistocene wolves, but there are no ancient data in that study. I guess the authors are actually citing Vila et al. 1999 as an example of coyote mitochondrial diversity (versus gray wolf diversity). In that

case I believe Sacks et al. 2021 - which some of the authors of the present manuscript also co-authored - is the most up-to-date summary of coyote mitochondrial diversity and, unlike Vila et al. 1999, they report many full mitochondrial genome sequences as opposed to only a 350bp fragment of CR.

Lines 252-253 "We found that dogs were not separated from wolves until at least 28 kya, as they share ancestry with all wolves postdating this time..."

While I eventually accepted the assertion made in the first part of this sentence, I didn't immediately understand the logic used to support it. Simply "sharing ancestry" alone is no evidence of anything. Specifically, I think the authors mean that dogs share drift with post-28 kya wolves that is not shared with pre-28 kya wolves (referring specifically to that shared shift along PC1), and hence the split post-dates that shared drift that appears to have occurred between 23-28 kya-ish. Could the authors please expand the explanation? This is an important point.

Line 278-279 "However, a model featuring the Siberian wolf and 10-20% basal ancestry does fit the Zhokhov dog ($p=0.29$)"

Is $p=0.29$ the correct p-value for this model? Fig. 1c seems to suggest it is <0.01 .

Line 284-285 "...European wolves behave like dogs..."

I know that "behaves like" is frequently used colloquially to mean "yields comparable results to" - I even use it myself in conversation with colleagues. But it does ascribe some agency to the data, and could be conflated with actual animal behaviour, so I think it would be preferable to use the latter phrasing above.

Line 293-295 "some Chinese wolves outcompete the 18ky old Siberian wolf"

As above for "behaves like". "Outcompete" is here being used to mean "provides a better fit" but the former could be misinterpreted as referring to ecological interactions between these animals. It would be less ambiguous to use the latter phrasing, in my opinion.

Figure S2

Some sample names in Figure S2 do not appear to correspond to the names listed in the sample metadata table (e.g. all of the samples with a TH prefix). For others, there appears to be some inconsistency between the predicted age in the plot (the teal distribution) and the sample metadata table (e.g. in Figure S2, SC19.MCJ010 appears to have a mean predicted age $>50,000$ years whereas in the sample metadata table its mtDNA date is given as 29,948 years, cf. the radiocarbon age at 29,943 calBP). Please clarify these issues.

Author Rebuttals to Initial Comments:

Referee expertise:

Referee #1: animal genomics

Referee #2: evolutionary genomics

Referee #3: palaeogenomics

Referees' comments:

Referee #1 (Remarks to the Author):

The manuscript "Grey wolf ancestry reveals a dual history of dogs" by Bergstrom et al analyzes a comprehensive repertoire of ancient DNA samples from the wolf and dog lineages to assess the evolutionary history of both species. The publication applies evolutionary analysis of genetic drift via f_3 and f_4 statistics. These show a temporal pattern of genetic drift suggesting regular admixture among populations in the last 100k years. In more recent times, the Siberian wolf population diverges to affect domestic dog ancestry in Europe. In North America, admixture with Coyote obfuscates simple analysis.

The study expands on (and somewhat contradicts) previous work by the same authors who previously suggested that dogs came from a single common wolf lineage (<https://www.science.org/doi/10.1126/science.aba9572>). The results are of interest to canine biologists and evolutionary biologists. The close affinity between domestic dogs and humans through history means that others may also be interested.

Data & methodology: There are a few aspects of the work that might be improved. The manuscript is well analyzed applying appropriate methods and strategies to form conclusions from the data. While the text reads well, and the figures have attractive colors, more care could be taken to help the non-evolutionary biologist understand what is depicted in the figures. Most of the data provided use appropriate statistics and error bars but the results - particularly in figures could be better explained (see comments following).

Conclusions: The publication makes use of expanded wolf ancestral data to inform canid evolutionary history and in particular the history of wolves. There remain some things to be explained, as at least one basal ancestor is yet to be determined in the future. Determining this ancestor is not possible with current resources.

References: Referencing is high quality.

We thank the reviewer for their positive assessment of the paper and their useful suggestions and comments, which we address below.

Suggested improvements:

First, the charts need to be reorganised to enable a single legend for multi-part panels (e.g. Extended Figure 1 a and b). The flipping between a y-axis of $-\log(P)$ and $\log(P)$ on chart axes is confusing to follow. In general, there is insufficient information in the Figure descriptions to understand what they depict. The numbering of supplementary figures in the text is all over the place and Figure S1 is not referred to at all.

We thank the reviewer for this and have increased the level of detail and clarity of the figure legends and supplementary figure referencing, as detailed further below.

All of the plots in Extended Data Figure 1 do actually have the same, shared legend, but the legend was included twice by mistake - this has been corrected, and we thank the reviewer for pointing this out.

Regarding the use of $-\log(P)$ and $\log(P)$ in chart axes, they are used for two different types of plots. We use $-\log(P)$ for the Manhattan plot (Fig 3a), as is standard in the field, aiming to highlight unusually small p-values at the top of the plot. We use $\log(P)$ for the display of *qpAdm* results in what we call "ocean plots" - these instead aim to highlight unusually large p-values at the top of the plot, which correspond to *qpAdm* models that can not be rejected. We therefore do not think it would be useful to switch around the signs for either of these two types of plots to match the other type.

Fig 1b legend should mention which samples (presumably wolves and dhole as no dog - but I do not see the dhole). Figure 1c, it is unclear which samples are represented in this chart - maybe give a number and refer to a supp table for description?

We have clarified the identities of the samples displayed in Figures 1a and 1b, including indicating what genomes were published previously, and we have listed the dogs that are included in 1c in the corresponding part of the Methods.

The explanation for Fig 4c should be clarified to help the novice reader interpret the chart (and is incorrect as it lists that the p-value is plotted rather than the $\log(P)$ -value)) and should be clear that strong significance refutes rather than supports the proposed ancestry. In Fig 4c the markers do not always seem to align properly in the columns for each dog. Is it intentional that North America is absent from the world map in Fig 4e? As a block the Figure 4 does not "stand alone".

We thank the reviewer for these useful suggestion and have addressed them as follows:

- Changed "p-values" to " $\log_{10}(\text{p-values})$ " in Fig. 4c legend
- Added "a low p-value means the model can be rejected" to the Fig. 4c legend to help reader interpretation of these not necessarily very intuitive results
- The reason the points in Fig 4c are not aligned in the columns is that they are randomly jittered to avoid overlap between them. We have clarified this in the legend by the addition of: "Points are jittered horizontally to avoid overlap".
- We have added North America to the map in Fig 4e.

Figure S1 is not referenced in the main text.

We thank the reviewer for noticing this, and have added references to figures S1 and S2 in the main text, after the sentence starting with “For the wolf samples without direct dates...”.

Figure S9 description "A positive value of this statistic implies that the non-canid species is closer to X, and a positive value implies it's closer to Siberia_Enygen.VAL_18A."
Is this correct? Should one say negative?

We thank the reviewer for spotting this typo, and have corrected the second “positive” to “negative”.

Main paper lines 153-4 - I am not certain from Figure S9 how this conclusion was reached?

In Figure S9, it can be observed that 1) African golden wolves have an elevated affinity to gray wolves in the Middle East and South Asia, and 2) other canid species appear slightly closer to the Siberian wolf used in these analyses than to some wolves from western China, e.g. Tibet, implying basal ancestry in the latter. These results are consistent with previously reported results cited in this sentence, but we do not feel like they warrant more attention in the main text. We do, however, discuss these results in further detail in the supplementary section containing this figure (“Relationships to coyotes and other non-wolf species”).

Lines 162-3 Doesnt Figure S9 (first panel) strongly suggest Coyote common ancestry?

The first panel of Figure S9 shows that for most ancient and modern wolves X, the coyote is symmetrically related to X and to the 100k-year old Siberian wolf Siberia_Enygen.VAL_18A. The primary exception to this are North American wolves, which the coyote is often closer to. We are not entirely sure what the reviewer means by ‘coyote common ancestry’, but we do discuss these results in further detail in the “Relationships to coyotes and other non-wolf species” section of the supplementary materials.

Figure S8 -please tell us explicitly that the up and down arrows in the titles relate to x and y axes respectively?

The arrows relate only to the values on the vertical axis (the observed statistics), and we have added the following to the figure legend to clarify this: “which relate to the values on the vertical axis”.

The description of "basal" ancestry, provided on lines 862-864 should be provided at least briefly earlier (e.g. before Line 87).

Following comments from Reviewer #3, in the revised version we no longer use the term “basal ancestry” in this context.

Clarity and context: The abstract provided is clear and outlines the major findings from the work.

Minor comments:

Fig 1b (not bold) line 119.

We thank the review for noticing this and have corrected it.

Referee #2 (Remarks to the Author):

Bergstrom et al. report findings of an immense undertaking, by way of sequencing 67 new ancient wolf genomes and an ancient dhole, and jointly analyzing these with previously published ancient and modern dog and wolf genomes. Their objectives are twofold: to more deeply understand the demographic history of wolves, and to hopefully use this information to gain new insight into the origin of dogs. The very much correct premise of this work is that analysis of ancient wolf genomes is crucial to understanding dog domestication, as more recent patterns of population expansion/contraction and admixture—both between wolf populations and between wolves and dogs—will obscure dog origins when one largely works with contemporary samples. The author's major findings are that populations structure in wolves during the Pleistocene was weaker than it is today; the majority of wolf ancestry in contemporary populations (including Europe and North America) originates from Siberia, but variable fractions of deep local ancestry persist outside of eastern Eurasia; the majority of dog ancestry originates from eastern Eurasia, but none of the populations represented by the ancient Siberian wolf samples appears to have been the direct ancestral source of dogs; and finally, that a minority fraction of dog ancestry originates from a western Eurasian source, either through admixture with local wolves or as the result of a second domestication process.

Broadly speaking, the presented work represents a significant advance in our understanding of wolf population history and dog domestication, ruling out a northern European origin for dogs and getting us geographically closer to the source population(s) from which dogs originated. But, in the absence of any definitive resolution regarding the geographic origin of dogs, I am on the fence as to whether the presented work represents a major advance in our understanding as to the origin of dogs.

Overall, the paper is clearly written and I have no major concerns with methods or inferences drawn from them, and what concerns/questions I have follow below.

We thank the reviewer for their positive assessment of the paper and their useful suggestions and comments, which we address below.

1. The selection analyses are the weakest part of the paper and detract from the story line around the main points. They are based upon single-SNP analyses—even if those SNPs are detected based upon a minimum signal in adjacent SNPs. In 2021, without functional tests or phenotypic associations, my understanding is that top tier journals such as Nature are disinclined to publish such results as they smack of storytelling. Too many studies have produced different selection hits solely by employing different methods. My opinion is that the selection results should be largely relegated to the supplementary information. I'd rather see Extended Data Figure 1 (or at least parts of it) moved to the main figures, replacing Figure 3, as it is central to the new findings on wolf demographic history.

The editor has requested that we keep the selection analyses in, and we do note that similar selection results without experimental follow-ups have been published by this journal recently (e.g. Mathieson et al. 2015, Mathieson et al. 2018, Margaryan et al. 2020 on ancient human genomes, Librado et al. 2021 on horses). We do agree with the reviewer though that the selection analyses needed expansion compared to other elements of the manuscript,

and in this revision we have performed substantial new analyses to flesh this out (as described further in response to reviewer #3). As we hope is now clear with extended analyses, the main selection signal is unusually well-supported--indeed no time series of a similar scope has been available before for any organism. However, we agree with the reviewer that the functional implications for the main region revealed to be under selection are unclear.

We thus have toned down the hypothesis that it was craniofacial adaptation specifically that drove the *IFT88* selective sweep, by 1) removing the reference to the fossil record on craniofacial diversity in wolves, 2) added the caveating sentence "But it is also possible that selection may have targeted unknown non-skeletal traits that *IFT88* variation may contribute to.", and 3) removed "craniofacial development gene" from before *IFT88* in the abstract.

We think that the the selection results are a significant advance in the study of Darwinian evolution, even if the function is not resolved, for the following reasons:

1) Direct study of selection through time at an unprecedented scale: Natural selection as conceptualised by Charles Darwin is ultimately genetic change through time. Most studies of natural selection in humans and other mammals use present-day genomes alone to indirectly detect the footprints of this historical process. Studying it directly is usually only possible in experimental populations of yeast or other small organisms with extremely short generation times. Some studies are now looking at genetic change through time in humans over a few millennia, on the order of 100 generations. The selection analysis in ancient wolves represents evolution over more than ~30,000 generations--100,000 years. In our view it is thus the first of a coming field of science to use ancient DNA to directly study Darwinian natural selection over large time scales.

It is true that selection has been inferred from ancient DNA before, primarily in humans, but those findings only concern local adaptation - certain variants increasing in frequency locally in e.g. Europe. What we are able to detect here is species-wide adaptation, with e.g. the *IFT88* variants going from close to 0% to 100% in the whole wolf species. These are thus variants that contribute to what makes wolves wolves. No such variants have been discovered through direct ancient DNA analyses in humans, or to our knowledge in any other macro-organism.

2) Enriching our understanding of wolf adaptation dynamics: Even if we cannot be sure of the function of the genes affected by the selective sweeps, the nature of these sweeps enrich our understanding of wolf history and adaptation dynamics. The results reveal that new adaptive variants were able to quickly reach the whole species, which affirms the model of high genetic connectivity, and perhaps also points to why the wolf species managed to be so successful in the Late Pleistocene. But at the same time, given that we only detect a handful of complete sweeps going from close to 0% to 100%, we know now that there were not hundreds of these - if there was more widespread adaptation across many different traits, it thus must have involved less dramatic (and perhaps polygenic) allele frequency changes. We have added two sentences to the Conclusions section making these points: *"Our finding that several selected alleles quickly reached fixation shows that adaptations spread to the whole interconnected population of Pleistocene wolves, a process that might have contributed to the survival of the species. At the same time, our results show that such*

rapid species-wide selective sweeps only occurred a few times over the last ~100,000 years.”

3) Functional information does exist for *IFT88*: We agree that the functional implication or the traits under selection could be non-skeletal traits. However, we do find the possible functional link noteworthy. If we had generated an *IFT88* knockout mouse for this paper, that would have been a state-of-the-art test of the function of this selected gene. But thanks to decades of work by the mouse genetics community this experiment has already been performed, allowing us to draw upon that functional information directly.

2. Figure 1a. It took me a minute to orient myself geographically since the figure is essentially looking down on the north pole. Perhaps some latitudinal lines would make this clearer or some other means of geographical orientation on the figure?

We thank the reviewer for this useful suggestion and have added a latitude and longitude grid to the map in Figure 1a.

3. In the paper there is reference to a continuous process of ancestry turnover but looking at Extended Data Figure 1b, and Figure S4—there does seem to be a shift over 10k years to largely ancestry originating from ancient Siberian wolf samples. This ancestry is a major point of the paper, and this not very continuous change is biologically interesting, so perhaps a shift in language is warranted along with at least some speculation as to what is driving what looks like a step function in the F_4 statistics.

The ancestry shift occurring around the last glacial maximum ~25 kya is indeed intriguing, but we do not necessarily believe that this is evidence of some kind of discontinuity or novel population dynamic. To further test this, we simulated a panmictic population with constant effective population size, and calculated the same type of f_4 -statistics as displayed in Figure S4 on this simulated data. The results (new Figure S5) show that the same ‘step function’ patterns in the f_4 -statistics are observed in this entirely static population simply moving forward in time. We outline some intuition for this in the supplementary material (page 9) - essentially, in a panmictic population, one will always be closer to a less distant-in-time ancestor than to a more distant-in-time ancestor. Our empirical data has a bit of a sampling gap between 23 and 28 kya, which might make the shift at this particular point in time appear slightly less gradual. But Figure S4 shows essentially the same patterns at earlier periods in the time series as well.

All in all, given how well the empirical patterns are replicated by the panmictic simulation, we do not think that there is evidence at present to suggest any unusual discontinuities in wolf population history in the period around 25 kya. To further clarify the interpretation of these results, including the new simulation results, we have rephrased the relevant sentences to: *“However, we also find that the same pattern is visible when contrasting affinities to younger versus older wolves at any point during the last 100 ky, not only at the LGM (Fig. S4). Using simulations, we confirmed that the observed temporal relationships are largely similar to what would be expected in a panmictic population (Fig. S5).”*

4. Lines 252-256 regarding timing of dog domestication. It is unclear how the statements wrt the separation of dogs and wolves are supported by either Fig. 1c or Extended Data Fig 1b. Also, I think the authors meant to say “We found that dogs were not separated from wolves until at least 28kya as they share ancestry with all wolves predating this time...” whereas they said “postdating.” I think there are other places in the manuscript where the authors language is unclear wrt whether they are looking forward or backward in time. That said, the shared drift plots can only make inferences on dog “separation” from wolves based upon the dates of the ancient dog samples, all of which postdate most genetic estimates of dog origins.

We agree that the phrasing of this sentence did not make it sufficiently clear what aspect of the results the claims on separation timing are based on. The rationale here - as also expressed by Reviewer #3 below - is that dogs share genetic drift with younger wolves (postdating 28 kya) which is not shared with older wolves (predating 28 kya). This means that dogs were still in genetic contact with wolves for some time after 28 kya. This observation can be seen in Fig. 1c, where dogs cluster with post 28-kya ancient and present-day wolves in the PCA. And more directly in Extended Data Figure 1b, where dogs share more drift with an 18k-year old wolf than with a 28k-year old wolf.

To clarify this, we have rephrased the sentence as follows: “We found that dogs share more genetic drift with wolves that lived after 28 kya than with those that lived before this time, which implies that the progenitors of dogs were genetically connected to other wolves at least until 28 kya.”

Additionally, as a further caveat for how there might not necessarily be a simple relationship between shared drift and the process of domestication, we have also added: “However, until the nature of the divergence process is better understood, it can not be ruled out that domestication had started before this point.”, as we agree with the referee comments that while our results allow a firm conclusion of multiple dog ancestries, they do not as strongly constrain the timing of dog domestication.

Regarding the estimation of dog-wolf divergence from MSMC2 plots based upon artificially constructed X diploids, there are two potential issues. First, the rapid increase in N_e deemed to be a signature of divergence could also be produced by other factors, namely admixture with another lineage, even if that admixture affected both populations from which the X chromosomes were drawn. It is easy to imagine a case where a) admixture with a third divergent lineage produces elevated heterozygosity and 2) the sampling of one X per population leads to each X including alternative alleles at the heterozygous sites produced by admixture. There truly is elevated heterozygosity in both target populations being analyzed but it has nothing to do with divergence. The strength of such an effect would be influenced by the population minor allele frequencies (at heterozygous sites) and the timing of admixture events. Second, it is challenging enough to hypothesize divergence events from MSMC plots comprised of true diploid individuals, let alone pseudohaploids comprised of low coverage genomes. Third, in their Supplementary Material, several of the MSMC plots indicate even older hypothesized dates of dog-wolf divergence. Overall, I think strong caveats need to be placed on the authors' claim of a more recent dog-wolf divergence.

We agree with the reviewer that there is considerable uncertainty surrounding the results on the timing of the dog-wolf divergence, especially the MSMC analyses, and have followed the reviewer's suggestion of introducing stronger caveats on these results. We have done this as follows:

- 1) As mentioned above, we have replaced the concluding sentence in this paragraph, which read "*This divergence estimate of at most 28 kya lowers the previous earliest bound (35 kya) on the time of domestication.*" with the caveat sentence "*However, until the nature of the divergence process is better understood, it can not be ruled out that domestication had started before this point.*"
- 2) The results on timing no longer have their own section heading, previously titled "*Dog origins: constraints on the timing of domestication*", but instead are covered at the beginning of the dog ancestry section, to reduce their emphasis
- 3) We have removed the timing results from the abstract, to further reduce their emphasis

5. Wrt the inferred second source of dog ancestry, I feel more confident in the results obtained from modeling ancestry contributions via qpWave, qpAdm and qpGraph than I do with the PCA results presented in Fig 4a. The effect in the PCA plot by way of the shift of some dogs towards the SW Asia is in most cases subtle, and the more dramatic cases being few. The crowding of points, and the inability to know which points represent which breeds makes interpretation even harder: perhaps a blow up of that part of PCA space where the dogs are and some breed labels would be helpful in this regard. Furthermore, one can't rule out that some of these shifts in PCA space aren't due to recent, geographically specific admixture between dogs and wolves as has been reported previously in other studies. If the ancestry modeling analyses were framed in the text as a way to overcome the potential confounding of signals in the PCA that would largely ameliorate the issue.

We have now added in the text, before the ancestry modelling of the second ancestry source, "Testing the PCA observations with explicit models", to improve the framing of the PCA and ancestry modelling sections.

However, because of the particular way in which this PCA is performed, we do not think that the cline among dogs is due to recent admixture from dogs into wolves. The input for this PCA is only f_4 -statistics of the form $f_4(X,A;B,C)$, where A, B, C are combinations of wolves that lived before 28 kya and therefore likely before dog domestication. The observation in the PCA that dogs differ amongst themselves in these statistics therefore means that they have different relationships to wolves that lived before the likely time of dog domestication. The PCA is thus conceptually very similar to the *qpWave* results using pre-domestication wolves, but allows for a visual representation of the dog asymmetries within the context of wolf diversity.

We have rephrased the sentence describing this PCA in the main text to hopefully better communicate how it was performed: *“To reduce the effects of gene flow since the emergence of dogs on our analyses, we performed a PCA on f_4 -statistics quantifying, for all wolves and dogs from the last 25 ky, relationships only to wolves living before 28 kya (i.e. pre-LGM), and found that dogs show relationship profiles similar to those of Siberian wolves from 23-13 kya.”*

We do agree that it would be useful with more detailed labelling of the dogs in the PCA. While it's not possible to fit all the labels, we have added labels for a number of key dogs of interest in the PCA in Extended Data Figure 2 (there is not enough space in the main text figure). In addition to this, there is Supplementary figure 20 which is a version of the PCA without wolves, and this includes labels for all dogs. We have also remade the PCA using modern wolf array genotypes to have more detailed labelling (Supplementary Figure 22).

Referee #3 (Remarks to the Author):

Overview

Bergström and colleagues present nuclear genomic data from a truly impressive number of ancient gray wolf specimens. This is now the most densely sampled temporal and geographical genomic transect for any non-human animal species - it beats the ancient sampling for both mammoths and horses. The framework it provides is compelling: by characterising the pre-domestication distribution of wolf diversity, the authors can more precisely identify the timing, location, and number of dog domestication events. Bergström and colleagues rigorously test a huge range of ancestry scenarios and are able to confidently reject a number of hypotheses, which greatly narrows the range of plausible origins for modern domestic dogs (and modern wolves too). The authors central claim - a dual ancestry for modern dogs - appears to be beyond reasonable doubt, as the most parsimonious explanation for the observed ancestry asymmetries observed in modern wolves and dogs. This study also outlines clear priorities for follow-up research, namely sampling of additional ancient wolves from central Eurasia and the contiguous US. I congratulate the authors on an outstanding piece of work.

I have no major issues with the authors' methodology or their interpretation of the results, with the exception of some reservations about the strength with which the authors assert that fixation of several alleles was caused by selection as opposed to neutral processes (see below). The rest of my feedback mainly centres on clarity of communication and accuracy of terminology. With corresponding revisions - which I have no doubt the authors can satisfactorily perform - I would be pleased to recommend this excellent study for publication.

We thank the reviewer for their positive assessment of the paper and their useful suggestions and comments, which we address below. We have substantially expanded the selection chapter of the paper, and now demonstrate with simulations that neutral processes would not give rise to the evidence of selective sweeps we observe (see details below).

General comments

I really like Extended Figure 3b (the plot of f_4 results for modern dogs against two ancient wolves versus the f_4 results for the same modern dogs against two ancient dogs) and I felt it was one of the clearest and most compelling illustrations of the whole manuscript's main message. I would strongly encourage the authors to move it to the main text. In my opinion it could easily replace Fig. 4a - I found the PCA difficult to interpret anyway because the dogs are coloured by their ancestry and not geographical origin. I have to admit I also didn't find Fig. 4b to be strictly necessary; since it's only text anyway the same information could easily just be summarised in the paragraph on Lines 304-311.

We do like Extended Figure 3b as well, and have followed the reviewers suggestion by moving it into main text Fig. 4b, and as the reviewer suggests replacing the previous Fig. 4b (*qpWave* results, which are described in the text anyway, as the reviewer notes).

The fixation of alleles in both IFT88 & the olfactory gene cluster on chr15 appears to have occurred post 28 kya (Lines 215-243, Fig. 3b). But in the paragraph immediately prior (Lines

202-212) the authors flag this time period as the onset of wolf population structure, resulting from some change in demographic processes (either decline in effective population size, reduction in gene flow, or possibly both). As a result, this is the exact time period when drift would be expected to disproportionately result in false positive signatures of selection using the method the authors employed (which does not correct for demography). They are certainly outliers and still interesting signals - I don't debate that - but I'm not personally convinced on the basis of the results presented that they must necessarily result from selection (versus simply neutral processes). I'd be more comfortable if the authors acknowledged or addressed this caveat in the main text. Overall, when compared to the other sections of this study - where the data are explored and scrutinised exhaustively with multiple analyses - this section on selection stood out to me as relatively bare-bones. So the authors might also consider omitting this section - thereby strengthening the central narrative about ancestry and origins - and instead follow up with a more in-depth study focused on adaptation (which I suspect they may already be planning anyway).

We agree with the reviewer that the section on selection was the least exhaustively investigated, and have now performed multiple additional analyses to confirm the robustness of the results.

New neutral simulations

Firstly, we have performed neutral simulations to address the concern that the observed peaks of major allele frequency change could have resulted from genetic drift alone, rather than natural selection. Across four neutral simulations with different effective population sizes (including an unlikely small size of 10,000), we observe no false positives when applying our selection scan. The large outlier peaks observed in the empirical data would thus be very unexpected under genetic drift alone, and therefore very likely reflect natural selection. The results from these analyses are described in detail in the new supplementary section "*Neutral simulations to assess the robustness of the selection scan*", and Supplementary Figure 17. We also have added QQ-plots comparing the empirical and simulated results to the main text Fig. 3b.

A major contributing factor to why this analysis is able to avoid false positives arising from genetic drift is the application of 'genomic control' (Devlin & Roeder 1999, *Biometrics*), which in this context aims to account for the genome-wide magnitude of the allele frequency variance introduced by drift that would otherwise lead to an inflation of p-values. In this way, the analysis does in practice account for demographic history, and genomic control has been used previously for the same purpose in state-of-the-art selection scans using ancient DNA (Mathieson et al. 2015, *Nature*; Mathieson et al, 2018, *Nature*). We did not highlight this use of genomic control in the submitted version of the manuscript, but have expanded on this in the new supplementary section and also added the following to the main text: "*and applying genomic control to correct for allele frequency variance introduced by genetic drift*", and the following sentence to the Methods section: "*The application of genomic control here aims to use the magnitude of temporal allele frequency variance observed genome-wide to account for what is observed from genetic drift alone given wolf demographic history.*"

Supplementary figure 17. Selection scans on simulated neutral populations. Manhattan plots of $-\log_{10}(p\text{-values})$ from selection scans performed on four simulated, panmictic populations with different effective population sizes (N_e). The λ value displayed for each plot is the inflation factor estimated through genomic control. Corresponding QQ plots are displayed to the right of each manhattan plot.

New genealogical selection scan with only modern genomes

Secondly, we have performed a separate scan for natural selection that uses only present-day wolf genomes, and which replicate *IFT88* as the most strongly outlying locus. Here, we used *Relate* (Speidel et al. 2019, *Nature Genetics*) to estimate genealogies and the Time to Most Recent Common Ancestor (TMRCA) in modern wolf genomes. The *IFT88* locus has the smallest inferred TMRCA in the entire genome of the present-day wolf population, and further analysis of the genealogies with *CLUES* (Stern et al. 2019, *PLoS Genetics*) estimates the TMRCA to $\sim 70,000$ years ago and a selection coefficient of $s=0.02$. This provides largely independent evidence using an orthogonal data set to the ancient wolf genome time series. We have added this analysis to the Supplementary Material, and main text Figure 3d.

Lastly, we have also:

- Added a sentence to the main text about the second strongest selective sweep. Interestingly, this sweep affects a region just 2.5 Mb downstream of the *IFT88* sweep (meaning it is not really visible in the genome-wide Manhattan plot), and the allele frequency trajectories imply selection during a similar time frame. This raises the possibility of correlated selection in these two nearby regions of the genome, perhaps if the second region is involved in long-range regulation of *IFT88*. While this is not something we can test, we think that the basic observation of two nearby, but independent, strongly selected peaks is worth mentioning.
- Slightly tweaked the p-value “neighbourhood filter” described in the Methods, which previously used the raw p-values but now more appropriately uses the genomic control corrected p-values, resulting in a more conservative list of 24 selection sweeps.
- Displayed allele frequency over time for the lead SNPs from all of the detected peaks, not just a few selected ones (Supplementary Figure 18).

Specific comments

Line 150 "...ancestry that is basal to the oldest Siberian wolves..."

TL;DR - Please revise usage of "basal" throughout.

The authors use "basal" here and throughout the manuscript, but the term is not defined for the reader until the end of the methods on Lines 864-865: "basal to (i.e. diverging early from, and lacking shared genetic drift with)". Even then, this definition differs from the meaning and usage of the word in phylogenetics, and risks perpetuating a persistent misconception (see summary by Krell & Cranston 2004; <https://doi.org/10.1111/j.0307-6970.2004.00262.x>). In brief, "basal" simply and strictly refers to a part of the tree that is closer to the root than another (i.e. older). Usually the term applies to nodes, but it can also apply to

taxa/lineages/populations (hereafter just taxa) but - importantly! - only when the representatives of those taxa are not coeval (e.g. a 20 kya wolf is more basal than a modern wolf, but two 20 kya wolves are equally basal). Therein lies the problem with the usage of basal in this paper, because introgressed ancestry does not derive from nodes, it derives from taxa. And admixture can only occur between coeval taxa. Because - as above - neither coeval taxon can be regarded as basal with respect to another (they are equally old, equally distant from their common ancestor), the ancestry inherited from both parties is equally basal. So - with the possible exception of time travel shenanigans - the concept of "basal ancestry" doesn't really make sense.

What the authors are actually referring to is ancestry from an unsampled or "ghost" lineage. They already specifically invoke this concept on Line 285-286 in reference to the ancestry of the Zhokov dog: "We therefore interpret the results for dogs as similarly reflecting some unsampled wolf ancestry..." But the "basal ancestry" referred to on Line 150 is also represented as "unsampled wolf ancestry" in Figure 1C, though - importantly - in these two cases the ancestry is coming from two different unsampled lineages: the former ancestry donor is a member of a sister lineage to ancient Siberian wolves, and the latter a sister lineage to all sampled modern and ancient wolves. It would be more accurate - and much clearer for the reader - if the authors entirely abandoned the use of "basal" and instead explicitly described the affinities of the unsampled lineage in each case. For example, the sentence beginning on Line 148 could instead read: "Most analysed present-day Eurasian wolves likely retain local Pleistocene ancestry, as they are best modelled by qpAdm as having 10-40% ancestry from an unsampled sister lineage to all modern and ancient wolves analysed in this study (Fig. S10, S11)." Where "basal ancestry" is used in the key for figures, it could easily be replaced by "unsampled ancestry", "ghost ancestry", or similar as appropriate.

We agree with the reviewer that the terminology is thorny when describing results of this nature, and have changed the terminology in the revised manuscript.

We would not object to their view on how the term "basal" should be applied within the context of phylogenetics. However, within the context of ancestry, the term "basal" has been used slightly differently, including in this journal (e.g. "Basal Eurasian ancestry", Lazaridis et al. 2014, *Nature*). In this usage, "basal" is always a relative term - a branch of ancestry is not inherently basal in itself, rather it is basal relative to other ancestries. For example, Neanderthal ancestry is basal to modern humans, and, symmetrically, modern human ancestry is basal to Neanderthals. The presence of basal ancestry is detectable specifically as a correlation to the allele frequencies of an outgroup.

This being said, in the revised manuscript we have changed this terminology, following the reviewers suggestion to avoid the term "basal ancestry" so as to avoid risks of confusion. We replace it with "unsampled ancestry" and slight variations thereof (e.g. "component to represent unsampled, divergent ancestry"). We have retained the usage of "basal" only when referring to the relationship of very old wolves to present-day wolves (e.g. line 87 in the original submission), as the temporal difference makes the phylogenetic definition outlined by the reviewer applicable. To also hopefully help interpretation of our results, we have added as Extended Data Figure 4 a schematic that summarises how deep, unsampled ancestry could underlie our observed qpAdm results.

Line 165-166 "...two Pleistocene Alaskan wolves also carry mitochondrial lineages that have been found in coyote"

There are six Pleistocene samples from Alaska in the dataset. Which two have the coyote-like mitochondrial haplotypes? And as a follow-up, are the the two ancient wolves with coyote haplotypes mentioned here the same individuals as the two non-modern coyotes mentioned on Line 757: "...included 3 coyotes as outgroups (1 modern, 1 14C dated and 1 with an infinite 14C date)"? If so, its not strictly correct to refer to them as coyotes in the second instance.

This might seem a trivial thing, but the ages for three of those six Alaskan wolf samples are inferred based only on the mtDNA analyses. The minimal representation of coyote mitochondria in the dataset could mean that the estimated ages for the two ancient samples with coyote-like haplotypes are biased (or at least imprecise). It's unlikely to be seriously problematic, but it would be useful to flag these two samples - perhaps in the sample metadata table - so that readers can easily cross-reference and judge for themselves.

The two Pleistocene wolves with coyote-like mitochondria are SC19_MCJ015 and SC19_MCJ017 (these are from the Yukon rather than Alaska - we have corrected this in the main text). They are indeed the same individuals mentioned on line 757, and we have rephrased this line to say "coyote-like sequences". We have also added the IDs of those two samples to the Methods section and expanded the relevant branch of the phylogeny in Supplementary figure 1 to show these samples too. We thank the reviewer for pointing out that this information was not included.

I also found the intent of the in-text citation on Line 166 a little ambiguous. I originally read it as meaning that Vila et al. 1999 had already identified these two mitochondrial haplotypes in Pleistocene wolves, but there are no ancient data in that study. I guess the authors are actually citing Vila et al. 1999 as an example of coyote mitochondrial diversity (versus gray wolf diversity). In that case I believe Sacks et al. 2021 - which some of the authors of the present manuscript also co-authored - is the most up-to-date summary of coyote mitochondrial diversity and, unlike Vila et al. 1999, they report many full mitochondrial genome sequences as opposed to only a 350bp fragment of CR.

We agree that this citation was ambiguous, and have removed it to avoid confusion. The observation described in this sentence is from our analyses, so a citation is not needed here.

Lines 252-253 "We found that dogs were not separated from wolves until at least 28 kya, as they share ancestry with all wolves postdating this time..."

While I eventually accepted the assertion made in the first part of this sentence, I didn't immediately understand the logic used to support it. Simply "sharing ancestry" alone is no evidence of anything. Specifically, I think the authors mean that dogs share drift with post-28 kya wolves that is not shared with pre-28 kya wolves (referring specifically to that shared shift along PC1), and hence the split post-dates that shared drift that appears to have occurred between 23-28 kya-ish. Could the authors please expand the explanation? This is an important point.

The reviewer is absolutely right, and the original phrasing was not sufficiently clear. We have rephrased the sentence as follows: "*We found that dogs share more genetic drift with wolves that lived after 28 kya than with those that lived before this time, which implies that the progenitors of dogs were genetically connected to other wolves at least until 28 kya.*" In the revised version we have also added a further caveat to this observation, as it does not necessarily strongly constrain the timing of dog domestication.

Line 278-279 "However, a model featuring the Siberian wolf and 10-20% basal ancestry does fit the Zhokhov dog ($p=0.29$)"

Is $p=0.29$ the correct p-value for this model? Fig. 1c seems to suggest it is <0.01 .

Yes, this is the correct p-value for this model. This is the point in the upper-right corner of Fig. 4c, which is above the $p=0.01$ dashed line and has a value of 0.29.

Line 284-285 "...European wolves behave like dogs..."

I know that "behaves like" is frequently used colloquially to mean "yields comparable results to" - I even use it myself in conversation with colleagues. But it does ascribe some agency to the data, and could be conflated with actual animal behaviour, so I think it would be preferable to use the latter phrasing above.

We agree with the reviewer that it's better to avoid this kind of language, and have rephrased the line to say "obtain results very similar to those of dogs".

Line 293-295 "some Chinese wolves outcompete the 18ky old Siberian wolf"

As above for "behaves like". "Outcompete" is here being used to mean "provides a better fit" but the former could be misinterpreted as referring to ecological interactions between these animals. It would be less ambiguous to use the latter phrasing, in my opinion.

We agree with the reviewer that it's better to avoid this kind of language, and have rephrased the line to say "provide a better fit than".

Figure S2

Some sample names in Figure S2 do not appear to correspond to the names listed in the sample metadata table (e.g. all of the samples with a TH prefix). For others, there appears to be some inconsistency between the predicted age in the plot (the teal distribution) and the sample metadata table (e.g. in Figure S2, SC19.MCJ010 appears to have a mean predicted age $>50,000$ years whereas in the sample metadata table its mtDNA date is given as 29,948 years, cf. the radiocarbon age at 29,943 calBP). Please clarify these issues.

Some of the mitochondrial genomes used for the mitochondrial tip dating were obtained from previously published studies, and so are not listed in the metadata table presenting the novel whole genomes produced in this study - we have tried to clarify this in the Methods description of these analyses. Regarding figure S2, what is displayed is not the results of the

mitochondrial tip-dating directly (those results are included in the metadata table), but rather results from a stand-alone follow-up experiment - we pretended that radiocarbon dated samples were undated, and tested how well we could predict their age through mitochondrial tip-dating. For SC19.MCJ010 and a couple of other samples, these predictions were not very close to the true radiocarbon dates. But those predictions are not used for any analyses, instead we of course use the radiocarbon

Reviewer Reports on the First Revision:

Referees' comments:

Referee #3 (Remarks to the Author):

Bergström and colleagues have thoroughly addressed all of my queries and requests for clarification. In particular, the new analyses that they have added (comparison with neutral simulations & application of Relate/CLUES to modern wolf genomes) and the expanded explanation of the "genomic control" principle employed in their original analyses make the natural selection section of the study much more rigorous and compelling. And I thank the authors for indulging my pedantry re: the term "basal".

As one final minor request: I wonder whether the authors would consider uploading their mitochondrial genome consensus sequences to GenBank? I note that the mitochondrial sequences could easily be reproduced using the sequencing data that the authors have already uploaded to ENA, but making them available through GenBank would allow easier re-use by researchers who do not have the required bioinformatics expertise or access to high-throughput computing infrastructure to do so. Just something to consider - either way, the authors' choice in this matter should not impact on editorial decisions.

Otherwise, I have no further substantive feedback for the authors and I am happy to recommend that this study be accepted for publication. I congratulate the authors on an impressive and important contribution.

Referee #4 (Remarks to the Author):

The authors have successfully addressed all the concerns raised by reviewer #2.

Point 1) The authors make an excellent argument to retain the selection analysis in this paper. The authors correctly argue that:

1) The allele frequency change in the IFT88 variant is indeed dramatic (going from a 0% frequency to a 100% frequency) which is a remarkable and robust result that argues in favor of very strong positive selection acting on this SNP. The fact that the TMRCA in this region is the smallest one in the genome provides a second compelling line of evidence in favor of strong positive selection acting in the region.

The authors also point out that the results in this paper enrich our understanding of wolf adaptation dynamics by showing that there are few hard sweeps in wolves in the past; and also argue correctly that there is functional information for IFT88 from knockout experiments on mice in this particular gene.

Point 2) Addressed.

Point 3) The simulations performed by the authors explain the "discontinuity" patterns seen in the f4

statistics when going backwards in time. The “discontinuity” pattern refers to the increase in f_4 values taking place around 30K years ago. The authors perform simulations under a constant population scenario and find that this “discontinuity” pattern is consistent with that demographic scenario without the necessity to invoke a more complex demography. The simulation results under a constant population size scenario backs up the authors’ claim that “homogenization due to connectivity thus seems to be driving Pleistocene wolf relationships.”

Point 4) The authors successfully address this comment correctly, and the edits make the relationships between dogs and the temporally sampled wolves more clear. Also, the edits regarding the timing of domestication based on the MSMC results are balanced and clear.

Point 5) The authors explain well the reasoning behind their PCA analysis, and their edits help the reader understand better how the PCA was performed. The edits on the Supplementary Figure also add the “breed labels” requested by the reviewer.

Overall, the authors have done an amazing job solving the reviewer #2’s comments. This paper represents a very valuable contribution to understand the evolution of wolves with solid new insights into their demographic history and how natural selection have driven dramatic allele frequency changes during the Pleistocene. This manuscript helps us get a better understanding of the origins of dogs which is a topic of high interest on ancient DNA studies and on the field of Evolutionary Biology.

Author Rebuttals to First Revision:

Referees' comments:

Referee #3 (Remarks to the Author):

Bergström and colleagues have thoroughly addressed all of my queries and requests for clarification. In particular, the new analyses that they have added (comparison with neutral simulations & application of Relate/CLUES to modern wolf genomes) and the expanded explanation of the "genomic control" principle employed in their original analyses make the natural selection section of the study much more rigorous and compelling. And I thank the authors for indulging my pedantry re: the term "basal".

As one final minor request: I wonder whether the authors would consider uploading their mitochondrial genome consensus sequences to GenBank? I note that the mitochondrial sequences could easily be reproduced using the sequencing data that the authors have already uploaded to ENA, but making them available through GenBank would allow easier re-use by researchers who do not have the required bioinformatics expertise or access to high-throughput computing infrastructure to do so. Just something to consider - either way, the authors' choice in this matter should not impact on editorial decisions.

Otherwise, I have no further substantive feedback for the authors and I am happy to recommend that this study be accepted for publication. I congratulate the authors on an impressive and important contribution.

We thank the reviewer for their constructive and useful suggestions, which have helped us make the paper stronger.

Following the reviewers final request, we are now providing the mitochondrial consensus sequences as a supplementary data file with the paper. (As these sequences are reference-guided rather than de-novo assembled, do not contain indels, and are of varying quality due their ancient origin, we think that GenBank is not the most appropriate place for them.)

Referee #4 (Remarks to the Author):

The authors have successfully addressed all the concerns raised by reviewer #2.

Point 1) The authors make an excellent argument to retain the selection analysis in this paper. The authors correctly argue that:

1) The allele frequency change in the IFT88 variant is indeed dramatic (going from a 0% frequency to a 100% frequency) which is a remarkable and robust result that argues in favor of very strong positive selection acting on this SNP. The fact that the TMRCA in this region is the smallest one in the genome provides a second compelling line of evidence in favor of strong positive selection acting in the region.

The authors also point out that the results in this paper enrich our understanding of wolf adaptation dynamics by showing that there are few hard sweeps in wolves in the past; and also argue correctly that there is functional information for IFT88 from knockout experiments on mice in this particular gene.

Point 2) Addressed.

Point 3) The simulations performed by the authors explain the “discontinuity” patterns seen in the f_4 statistics when going backwards in time. The “discontinuity” pattern refers to the increase in f_4 values taking place around 30K years ago. The authors perform simulations under a constant population scenario and find that this “discontinuity” pattern is consistent with that demographic scenario without the necessity to invoke a more complex demography. The simulation results under a constant population size scenario backs up the authors’ claim that “homogenization due to connectivity thus seems to be driving Pleistocene wolf relationships.”

Point 4) The authors successfully address this comment correctly, and the edits make the relationships between dogs and the temporally sampled wolves more clear. Also, the edits regarding the timing of domestication based on the MSMC results are balanced and clear.

Point 5) The authors explain well the reasoning behind their PCA analysis, and their edits help the reader understand better how the PCA was performed. The edits on the Supplementary Figure also add the “breed labels” requested by the reviewer.

Overall, the authors have done an amazing job solving the reviewer #2’s comments. This paper represents a very valuable contribution to understand the evolution of wolves with solid new insights into their demographic history and how natural selection have driven dramatic allele frequency changes during the Pleistocene. This manuscript helps us get a better understanding of the origins of dogs which is a topic of high interest on ancient DNA studies and on the field of Evolutionary Biology.

We thank the reviewer for their positive assessment of our response to the comments and suggestions from reviewer #2, and for the positive assessment of the paper overall.